

# Dynamics of China's Forest Carbon Storage: The First 30 m Annual Aboveground Biomass Mapping from 1985 to 2023

Yaotong Cai [1], Peng Zhu [2], Xing Li [1], Xiaoping Liu [1], Yuhe Chen [1], Qianhui Shen [1], Xiaocong Xu [1], Honghui Zhang [3], Sheng Nie [4,5], Cheng Wang [4,5], Jia Wang [6,7], Bingjie Li [1], Changjiang Wu [1], Haoming Zhuang [1]

[1]School of Geography and Planning, Sun Yat-Sen University, Guangzhou, 510275, China
[2]Department of Geography, The University of Hong Kong, Hong Kong SAR, 999077, China
[3]Guangdong Engineering Center for Intelligent Spatial Planning, Guangdong Guodi Planning Science Technology Co. Ltd, Guangzhou, 510650, China
[4]International Research Center of Big Data for Sustainable Development Goals, Beijing 100094, China
[5]Key Laboratory of Digital Earth Science, Aerospace Information Research Institute, Chinese Academy of Sciences, Beijing 100094, China
[6]Beijing Key Laboratory of Precision Forestry, Beijing Forestry University, Beijing, 100094, China
[7]Ministry of Education of Engineering Research Center for Forest and Grassland Carbon Sequestration, Beijing Forestry
University, Beijing, 100094, China

*Correspondence to*: Xiaoping Liu (liuxp3@mail.sysu.edu.cn)

**Abstract.** Accurate estimation and monitoring of forest aboveground biomass (AGB) are essential for understanding carbon
dynamics, managing forest resources, and guiding environmental policies. However, the spatial and temporal patterns,
dynamics, and driving factors of forest AGB in China over recent decades remain insufficiently understood, hindering
ecosystem analysis and forest management strategies. This study combines multi-source remote sensing data with residual
neural networks (ResNets) to develop the first 30 m resolution annual China Forest AGB dataset (1985–2023) with uncertainty
quantification. Validation results confirm the robustness of the ResNets model, achieving an $R^2$ of 0.92, RMSE of 16.06 Mg/ha,
and Bias of 0.06 Mg/ha against GEDI footprint AGBD, and an $R^2$ of 0.63, RMSE of 68.26 Mg/ha, and Bias of -19.87 Mg/ha
against independent multi-year ground survey data. The dataset reveals a notable increase in China's average forest
aboveground biomass density (AGBD) from 95.74±11.30 Mg/ha in 1985 to 122.69±13.94 Mg/ha in 2023. During this period,
total forest aboveground carbon (AGC) stock rose from 5.50±0.23 PgC to 13.97±0.87 PgC, establishing China's forests as a
significant carbon sink over the past four decades, with a net carbon sink of 0.22±0.01 PgC yr⁻¹, offsetting 11.5%–14.9% of
China's fossil fuel and industrial emissions. Forest growth contributed 65.1% (5.75 PgC) of the total AGC increase, while
forest expansion accounted for 34.9% (3.09 PgC). This dataset provides critical information for forest carbon accounting in
China and offers valuable insights for climate change mitigation, ecosystem conservation, and sustainable land management.

## 1 Introduction

Forests cover 31% of the Earth's land surface and play a vital role in maintaining ecological balance (Houghton, 2005),





supporting biodiversity and delivering essential ecosystem services (Brockerhoff et al., 2017). Over the past three decades,

forests have absorbed an estimated 106.9 PgC, with a net carbon sink rate of 1.3 PgC yr$^{-1}$—equivalent to 91% of the terrestrial net carbon sink (Pan et al., 2024). This substantial carbon sequestration offsets approximately 17% of global fossil fuel emissions (Pan et al., 2024), positioning forest conservation and restoration as critical components of nature-based solutions (NBS) to address climate change and achieve carbon neutrality (Seddon et al., 2021).

Aboveground biomass (AGB) serves as a direct indicator of forest carbon stocks and dynamics, reflecting the capacity

of forests to act as carbon sinks (Houghton, 2005; Pan et al., 2011). As such, it is recognized as one of the 54 essential climate variables (ECVs) by the Global Climate Observing System (GCOS) (GCOS, 2017; Cartus et al., 2014; Quegan et al., 2019). Estimating forest AGB is essential for quantifying emissions from deforestation and degradation, directly supporting policy frameworks such as REDD+, the Paris Agreement, and the United Nations Sustainable Development Goals (SDGs) (Baccini, 2012; Xu et al., 2021). Moreover, understanding forest biomass stocks enhances insights into carbon storage patterns and

dynamics, which are critical for improving Earth system models (Bar-On et al., 2018; Fang et al., 2024; Feng et al., 2021; Friedlingstein et al., 2023; Liu et al., 2015).

China ranks fifth globally in forest area and boasts the largest area of planted forests in the world (FAO, 2020) (see Text S1 for information details of study area). Since the 1970s, the Chinese government has implemented several large-scale forestry programs, including the Three-North Shelter Forest Program, the Yangtze River and Zhujiang River Shelter Forest Projects,

the Natural Forest Protection Project, the Grain for Green Program, and the Beijing–Tianjin Sand Source Control Project, to enhance ecological quality and mitigate environmental degradation (Liu et al., 2008; Lu et al., 2018). These initiatives have driven substantial afforestation and reforestation efforts, increasing forest coverage from 12.98% in the 1980s to 24.02% in 2021, with the total forest area nearly doubling from approximately 1.15 million km² to 2.31 million km² (Fig. S1) (National Forestry and Grassland Administration, 2019; Cai et al., 2024).A recent study revealed that between 1990 and 2019, the carbon

sink capacity of global temperate forests increased by 30.2%, with China's afforestation and reforestation programs playing a critical role in these trends (Pan et al., 2024). While these efforts have undoubtedly enhanced global carbon sequestration, their contribution to AGB growth—particularly the spatial and temporal dynamics of carbon accumulation across China's forests—remains not fully quantified. This quantification is especially significant given that many agencies need to track AGB estimates over time to support REDD+ initiatives and fulfil national reporting obligations (Wulder et al., 2020). A systematic evaluation

of AGB changes in China over the past decades is therefore necessary to disentangle the impacts of afforestation policies, assess regional variations in carbon sequestration, and inform the next generation of land management strategies.

To achieve this goal, high-spatiotemporal-resolution, long-term assessments of forest AGB are critical for countries like China, which have implemented extensive afforestation programs. While mature forests exhibit gradual annual biomass changes that may not be detectable through yearly remote sensing observations, monitoring intervals of 3–5 years can

effectively capture meaningful growth patterns (Le Toan et al., 2011). In contrast, young, fast-growing forests or recently disturbed areas demand more frequent (e.g., annual) monitoring due to rapid carbon turnover in dynamic carbon pools, making such assessments crucial for accurately detecting disturbances and recovery phases (Quegan et al., 2019).



Traditional forest inventory methods, based on permanent sample plots and periodic (e.g., five-year) surveys, are widely used at the national level to estimate forest biomass and its dynamics. While these surveys provide valuable statistics, they are labour-intensive, lack spatial and temporal continuity, and are unsuitable for producing end-to-end, spatiotemporally consistent biomass estimations. Process-based ecosystem models (e.g., Dynamic Global Vegetation Model, DGVM), which simulate biogeochemical processes such as photosynthesis, carbon allocation, and uptake, often incorporate biotic, climatic, and anthropogenic drivers (Sitch et al., 2008). However, these models are limited by uncertainties in input datasets (e.g., climate and soil data), coarse spatial resolutions, and assumptions of stand homogeneity, resulting in significant uncertainties in biomass estimates and an inability to capture spatial variability in forest biomass. By contrast, remote-sensing-based models offer higher accuracy and are better suited for monitoring forest biomass dynamics, emerging as a promising alternative (Lu et al., 2016).

Spaceborne LiDAR systems (e.g., GEDI) provide accurate footprint-level forest AGB estimates but are limited in spatial and temporal coverage, making them unsuitable for nationwide, long-term monitoring. (Coops et al., 2021; H. Nguyen et al., 2019) While synthetic aperture radar (SAR) data enable spatially continuous monitoring, historical satellite missions like ALOS-2 and Sentinel-1 lack sufficient temporal depth to capture the multi-decade impacts of large-scale afforestation. Vegetation optical depth (VOD) derived from SAR has demonstrated strong correlations with AGB over long timescales (Fang et al., 2024; Moesinger et al., 2020) but suffers from coarse spatial resolution, limiting its effectiveness in capturing fine-scale AGB changes caused by deforestation, degradation, and recovery.

Currently, large-scale, long-term forest biomass monitoring systems primarily rely on multispectral satellite imagery (e.g., Landsat time series) combined with field inventory data or airborne LiDAR to estimate AGB (Coops et al., 2021). Efforts to extend the temporal coverage of AGB estimations generally employ one of three approaches: (1) Multitemporal prediction: constructing temporally generalizable models using multitemporal references to predict historical AGB based on spectral data and trends (Harris et al., 2021; Matasci et al., 2018; Wulder et al., 2020). (2) "Space-for-time substitution": Generating AGB baselines and modelling historical AGB changes using related forest parameters (e.g., tree cover), though this heavily depends on the robustness of parameter-AGB relationships and parameter accuracy (Chen et al., 2023). (3) Continuous estimation: Utilizing spectral segmentation algorithms (e.g., CCDC, LandTrendr) to derive AGB time series aligned with Landsat data, though this requires high computational resources and extensive processing, making it less feasible for large-scale applications (Fu et al., 2024). Among these, temporal prediction using annual Landsat time series offers an optimal balance for tracking forest changes, capturing abrupt and gradual transitions with manageable computational demands (Nguyen et al., 2019).

In China, achieving high-quality, high-resolution, and long-term monitoring of AGB remains a significant challenge (Chen et al., 2023; Huang et al., 2019; Zhang et al., 2019). One major limitation is the absence of forest-specific thematic maps that can reflect China's forest dynamics. Forest distributions derived from land-use/land-cover datasets often fail to accurately capture the scale of afforestation efforts (Cai et al., 2024), causing past datasets to overlook these dynamics and underestimate China's carbon sink capacity (Wang et al., 2020; Yu et al., 2022). Furthermore, there is a marked lack of large-scale, geographically diverse biomass survey data that encompass China's varied forest types, soil conditions, climatic zones, and

biomass gradients. Such data are crucial for developing accurate and generalizable models with strong spatial and temporal applicability. Although several static AGB maps have been produced for China using different methodologies, significant discrepancies persist (see Table S1). For example, forest AGBD estimates range from 57.05 Mg/ha (Santoro et al., 2021) to

160.74 Mg/ha (Hu et al., 2016), while estimates of aboveground carbon (AGC) stocks vary from 5.04 PgC to 11.06 PgC. These inconsistencies stem from differences in data collection periods, input data accuracy, and estimation approaches. A recurring issue in many AGB models is their reliance on field data that were initially intended for modelling purposes rather than for direct alignment with remote sensing observations. This temporal and spatial misalignment hampers the ability of models to effectively capture forest dynamics, reducing their reliability for national-scale carbon stock assessments (Araza et al., 2022;

Babcock et al., 2016; Le Toan et al., 2011).

In response to the urgent need for improved national biomass mapping and the lack of a comprehensive dataset to support continuous monitoring of AGB spatial dynamics, this study proposes a bottom-up spatial framework to quantify changes in China's forest biomass over the past four decades (1985–2023). This framework integrates a ResNet-based deep learning algorithm, leveraging over 50,000 multi-temporal GEDI AGB training samples (2019–2021) and remote sensing observations

to train the model and generate accurate AGB time series, thereby reducing uncertainties in inventory change estimates. The primary goals of this study are: (1) to generate the first 30 m resolution annual time series dataset of China's forest AGB from 1985 to 2023, referred to as the China Forest AGB Time Series Dataset (CFATD); and (2) to monitor and quantify long-term and short-cycle changes in China's terrestrial forest biomass, improving estimates of carbon sources (e.g., from deforestation) and carbon sinks (e.g., from forest regeneration and afforestation). By achieving these objectives, this study seeks to answer

four key questions: (1) What are the spatiotemporal trends of China's AGB carbon sink? (2) How much carbon have forests sequestered under China's large-scale afforestation initiatives? (3) How much of this sequestration is directly attributable to afforestation? (4) To what extent can forests contribute to China's carbon neutrality goals, and how much of the nation's anthropogenic carbon emissions have past forest carbon sinks offset? The resulting dataset provides a robust, data-driven foundation for understanding forest biomass changes and their implications for environmental policy, climate change

mitigation, and sustainable forest management.

## 2 Materials and methods

### 2.1 Remote sensing data

#### 2.1.1 Landsat imagery

This study utilized Landsat satellite imagery from the Google Earth Engine (GEE) platform, covering the period from

1985 to 2023. The dataset includes images from the Landsat 4/5 Thematic Mapper (TM), Landsat 7 Enhanced Thematic Mapper Plus (ETM+), and Landsat 8/9 Operational Land Imager (OLI) sensors, all acquired from Collection 2, Level 2, Tier 1 surface reflectance products, with a 30-meter spatial resolution. These products have undergone comprehensive atmospheric



and orthorectification corrections, with surface reflectance derived using the Land Surface Reflectance Code (LaSRC) (Wulder et al., 2022).

The Tier 1 dataset achieves high geographic accuracy, with a root mean square error (RMSE) of less than 12 meters, suitable for reliable time-series analyses and change detection. This accuracy tolerance is beneficial, as it is less than half the 30-meter pixel size of the multispectral bands, supporting precise environmental monitoring. Collection 2 offers several advancements over Collection 1, including enhanced geolocation accuracy, improved radiometric calibration, and a higher-accuracy digital elevation model (DEM) for orthorectification (Crawford et al., 2023).

Imagery is packaged into overlapping scenes using a standardized reference grid, each covering approximately 170 km by 183 km. To maintain consistency across time-series analyses, we used the same raw data source as Collection 2, Tier 1 products for composite generation. Each image includes three visible bands (red, green, blue), one near-infrared (NIR) band, and two shortwave infrared (SWIR) bands (see Table 1), along with a QA band for cloud and shadow masking (Zhu et al., 2015).

### 2.1.2 China annual tree cover dataset

The China Annual Tree Cover Dataset (CATCD) is the first long-term dataset capturing annual tree cover changes across China at a 30-m spatial resolution from 1985 to 2023 (Cai et al., 2024a). Developed using Landsat time-series imagery and advanced random forest-based ensemble learning techniques, the dataset has been rigorously validated against multiple reference datasets, achieving correlation coefficients of 0.70 to 0.96 and RMSE values ranging from 5.6% to 25.2%. The

dataset aligns closely with National Forest Inventory (NFI) trends, effectively capturing forest area dynamics driven by major afforestation and reforestation initiatives. This capability enables the development of an AGB dataset that reflects temporal forest changes with high fidelity. In accordance with the technical standards of the National Forestry and Grassland Administration (National Forestry and Grassland Administration, 2019) and forest definition in the previous study (Fang et al., 2001; Piao et al., 2009), a 20% tree cover threshold was applied to delineate forested areas, which formed the foundational

layer for AGBD estimation. Furthermore, leveraging the strong correlation between canopy cover and forest biomass (Chen et al., 2023), tree cover was also used as an explanatory variable in AGBD modelling.

### 2.1.3 Auxiliary data

        To enhance the accuracy of AGBD predictions by capturing the environmental and spatial factors influencing forest biomass, we integrated climate, topography, and geographic location as explanatory variables in our AGBD prediction model

(Su et al., 2016; Zhang et al., 2019). Climate variables were sourced from WorldClim 2.1, which provides global climate data for the period 1970–2000 at high spatial resolutions (~1000m). This dataset includes variables such as temperature, precipitation, and other bioclimatic factors. We selected four climate variables from WorldClim 2.1 that are relevant to forest growth: bio1 (annual mean temperature), bio4 (temperature seasonality), bio12 (annual mean precipitation), and bio15 (precipitation seasonality). To match the spatial resolution of other datasets such as Landsat, we resampled the climate data to





30m resolution using bilinear interpolation. Topographic factors were derived from the SRTM DEM V3, which provides 30m resolution elevation data. Based on the DEM data, we calculated slope and aspect using the GEE platform. Additionally, we created a 30m resolution geographic location image that includes two bands representing the latitude and longitude of the centre point of each 30m pixel.

## 2.2 Reference data

### 2.2.1 GEDI footprint AGBD product

The GEDI instrument, mounted on the International Space Station (ISS), uses three lasers to produce eight ground tracks of laser pulses, generating footprints approximately 25 meters in diameter with 60-meter spacing along the orbital path. Footprint-level AGBD is derived from models developed by collocating field-based AGBD estimates with simulated GEDI waveforms generated from discrete-return airborne lidar (Duncanson et al., 2022). The modeling approach stratifies candidates
by global regions and plant functional types (PFTs), using diverse functional forms to enhance prediction accuracy. Despite limited regional representation in GEDI's field-sampled biomass density data, the Level 4A (L4A) algorithm addresses geographic transferability, enabling predictions beyond the geographic range of training data (Kellner et al., 2023). Moreover, the GEDI dataset provides uncertainty estimates for each AGBD prediction, increasing its applicability in underrepresented regions.

We obtained 25-meter resolution AGBD predictions for 2019–2023 from the GEDI L4A Version 2.1 dataset via the Google Earth Engine platform. These data were used as both training and testing inputs for our ResNet model, which estimates forest biomass from time-series optical satellite observations spanning 1985–2023. To ensure data quality, we applied stringent filtering criteria, excluding samples flagged as low quality (l4_quality_flag = 0 or degrade_flag > 0), those with relative errors exceeding 50%, and measurements from areas with slopes greater than 30 degrees. Observations collected outside the growing
season were also excluded to maintain consistency with canopy conditions captured in Landsat imagery. After filtering, we retained 72,150 valid GEDI AGBD samples distributed across China's seven major ecoregions (Fig. S2 and Table S2). To ensure compatibility with the 30-meter resolution Landsat composites and other explanatory variables, the GEDI AGBD footprints were resampled to 30-meter resolution using bilinear interpolation.

### 2.2.2 Field survey data

To independently evaluate the accuracy of forest AGBD estimates from remote sensing, we compiled ground-based measurements from 4,243 forest plots across the study area using published literature(Fig. S3) (Avitabile et al., 2016; Luo et al., 2014; Usoltsev and Усольцев, 2020; Zhang et al., 2019). AGBD for each plot was calculated from stand variables (tree height, DBH) using allometric equations, providing a robust benchmark for validation. After quality control—removing inconsistent records, excluding plots larger than 1 hectare, and aligning survey years with publication dates when necessary—
2,109 plots were retained. These plots, surveyed between 1978 and 2008, represent diverse forest types (primary, secondary,





plantations) and dominant species such as *Betula platyphylla*, *Cunninghamia lanceolata*, and *Pinus koraiensis*. Plots surveyed prior to 1985 were used to validate the 1985 AGBD map.

### 2.2.3 Provincial statistics from national forest inventories

The National Forest Inventories (NFIs) provide provincial-level forest area and total forest volume data but lack biomass-
specific information for direct validation. Here, we utilized provincial forest AGC statistics derived from Fang et al., (2001), based on forest volume data from five NFIs (3rd NFI: 1984–1988 to 7th NFI: 2004–2008). These AGC estimates were obtained by converting the NFI-reported forest volumes into biomass using continuous biomass expansion factors (BEFs) (Fang et al., 2005). The reliability of the AGC estimates has been validated in previous studies (Fang et al., 2001).

### 2.3 Forest AGBD mapping using ResNet

#### 2.3.1 Model development

To improve computational efficiency and remove redundant information, we applied Recursive Feature Elimination (RFE) combined with five-fold cross-validation to identify the optimal set of features for forest AGBD mapping (Cai et al., 2020). The importance of the selected explanatory variables was further evaluated using the Random Forest method. Subsequently, these RFE-selected features were utilized as explanatory variables in developing the forest AGBD estimation model based on
the Residual Neural Network (ResNet).

Figure 1 illustrates the architecture of the ResNet used for AGBD estimation. Based on the original ResNet architecture (He et al., 2016), all 2D layers were replaced with their 1D counterparts. The model comprises two residual blocks, each consisting of a convolutional layer followed by batch normalization and a Rectified Linear Unit (ReLU) activation function. Each block adapts a skip connection mechanism that adds the block's input to its final activation map. This mechanism not
only mitigates the vanishing gradient problem in deep neural networks but also simplifies the task of learning residual corrections to the input for each block. The residual connection, which fuses the original inputs and convolutional features, allows the model to better capture data patterns by combining shallow and deep features. Following each residual block, max-pooling layers reduce the intermediate feature representations by half, thus decreasing memory consumption and accelerating network inference. Additionally, downsampling gradually increases the receptive field of the learned features, enabling deeper
layers to cover larger segments of the input data and encode more contextual evidence. To enhance model's generalization ability, a dropout layer with a rate of 0.5 is employed before the final fully connected layer consisting of two linear layers activated by ReLU function.

We randomly selected 80% of the multi-temporal GEDI AGBD samples (n=57,720) as the training set, while the remaining 20% (n=14,430) were reserved for multi-year validation to test the robustness of the model over time. The training
data were aligned with the observation dates of Landsat satellites to develop a temporally consistent deep learning model, which was then applied to the entire time series of images to generate the forest biomass time series dataset. During model





training, we adopted the Adam optimizer, an adaptive version of stochastic gradient descent. In each iteration of the optimizer, the loss was calculated based on random batches of 64 data samples (batch size=64). Based on this loss, the partial derivatives (gradients) of all model parameters were determined by backpropagation, using the chain rule to compute numerical gradients.

The model parameters were updated in small steps along the negative gradient direction to minimize the loss, with the step size controlled by a hyperparameter known as the learning rate, initialized to 0.001. The model was trained for 1,000 epochs, where an epoch refers to the entire training data set going through a complete process of forward propagation and back propagation.

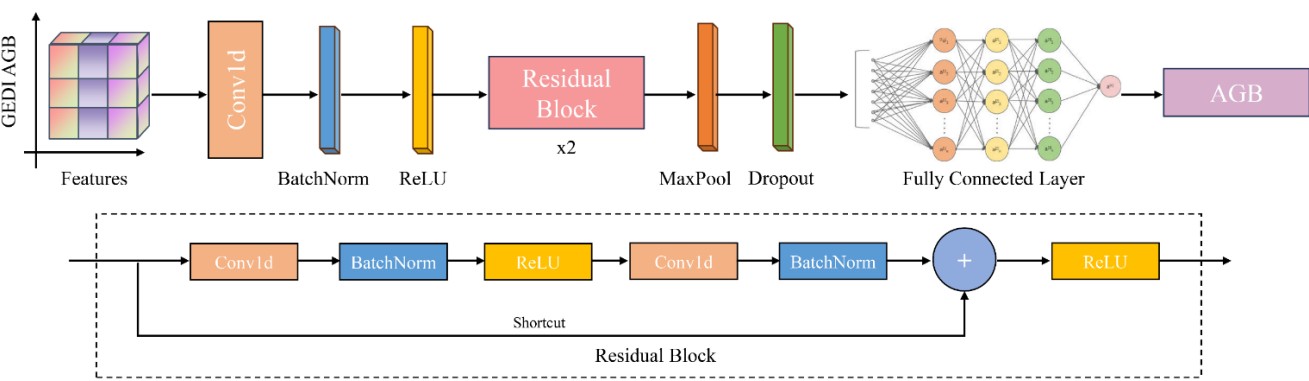

**Figure. 1:** Architecture of the ResNets used for AGBD estimation.

**2.3.2 Uncertainty estimation**

In this study, we employed quantile regression to quantify the uncertainty of AGBD estimates (Chung and Neiswanger, 2021; Shendryk, 2022). Assuming a Gaussian error distribution, the 95% prediction interval was approximately within ±2 standard deviations (SD), and the 68% prediction interval was approximately within ±1 SD of the AGBD estimates. Therefore,

we estimated the AGBD values at the 2.5th percentile and 97.5th percentile for each pixel using the ResNet model with quantile loss function (Eq. (1)). The difference between the quantile loss estimates at these percentiles provided the 95% prediction interval, allowing us to quantify the uncertainty (±1 SD) of AGBD estimates for each pixel. The uncertainty derived from the quantile regression can capture the variability in predictions associated with the model, reflecting modelling uncertainty derived from the spread of prediction intervals.

$$\begin{cases} L_q(y, y) = q(y - y)_+ + (1-q)(y - y)_+ \\ (\cdot)_+ = \max(0, \cdot) \end{cases} \tag{1}$$

where $L_q$ represents the quantile loss function, $q$ is the number of quantiles of interest, $y$ and $\hat{y}$ signify the observation and prediction, respectively. In this study, $q$ was set to 2.5% and 97.5% to obtain AGBD estimates at the 2.5th and 97.5th percentiles, respectively.





## 2.4 Accuracy assessment

For assessing the accuracy of the AGBD estimation model, we utilized validation metrics such as the coefficient of determination ($R^2$, Eq. (2)), root mean square error (RMSE, Eq. (3)), and bias (Eq. (4)). These metrics quantified the agreement between predicted AGBD values and reference values, providing insights into prediction accuracy and systematic errors.

$$R^2 = 1 - \frac{(n-1)\sum_{i=1}^{n}(y_i - y_i)^2}{(n-2)\sum_{i=1}^{n}(y_i - y_i)^2} \qquad (2)$$

$$RMSE = \sqrt{\frac{\sum_{i=1}^{n}(y_i - y_i)^2}{n}} \qquad (3)$$

$$Bias = \frac{\sum_{i=1}^{n}(y_i - y_i)}{n} \qquad (4)$$

where $n$ is the number of reference samples, $\hat{y}_i$ represents the estimated AGBD, and $y_i$ denotes the reference AGBD.

## 2.5 Comparison of existing AGB products

To further validate the effectiveness of our models, we compared the results against several external AGB datasets, focusing on both spatial and temporal dynamics (see Text S2 for details). For spatial comparison, we used static AGB datasets from global (e.g., Zarin map (Zarin et al., (2016)), Santoro map (Santoro et al., (2021))) and regional (e.g., Hu map (Hu et al., (2016)), Yang map (Yang et al., (2023))) sources to assess the spatial distribution and accuracy of AGBD estimates at different resolutions (from 30 m to 1000 m). For dynamic comparison, we utilized annual and multi-year datasets (e.g., Hengeveld map (Hengeveld et al., 2015), Liu map (Liu et al., 2015), Chen map (Chen et al., 2023), ESA CCI) spanning different periods from 260    1950 to 2021. This allowed us to evaluate how well the models captured forest biomass changes over time across various temporal scales and spatial extents. By comparing static and dynamic AGBD estimates, we provide a comprehensive view of the performance and robustness of the developed models under different conditions.

## 2.6 Forest AGC stock and change analysis

The AGB (unit: Mg) for each pixel was calculated by multiplying the estimated AGBD by the forest area derived from the CATCD for the corresponding year (represented as the product of forest cover fraction and pixel area). Carbon content was then derived from the biomass using a conversion factor of 0.5, allowing for the calculation of forest AGC for each pixel. The net annual change in forest carbon was calculated by aggregating the pixel-level changes.



**2.7 Quantification the impact of forest changes on AGC stock**

Using annual forest cover data from the CATCD, we analyzed forest dynamics from 1985 to 2023, focusing on forest
gain and loss. Forest gain includes both forest growth and expansion. Forest growth refers to areas that remained forested from
1985 to 2023 but experienced increased tree cover, while forest expansion refers to areas that transitioned from non-forest
(tree cover < 20%) to forest (tree cover ≥ 20%) during this period. On the other hand, forest loss comprises tree cover loss and
areas transitioning from forest to non-forest. Tree cover loss denotes areas that remained forested over time but showed a
reduction in tree cover, whereas the transition from forest to non-forest represents a significant reduction in forest land.
275    To quantify the impact of these forest dynamics on AGC, we categorized AGC changes that did not alter the land-use
type as tree cover change (TCC)-induced changes. This includes forest growth and tree cover loss. Changes that involved land-
use type alterations were classified as land-use and land-cover change (LULCC)-induced changes, encompassing forest
expansion and transitions from forest to non-forest. By integrating these dynamics with AGC estimates for 1985 and 2023
(calculated in Section 3.4), we quantified the contribution of forest gain and loss to the overall AGC changes, determining the
280    percentage contribution of each dynamic to the fluctuations in carbon stocks.

**3 Results**

**3.1 Feature selection and importance**

The RFE results indicated that the model's cross-validation score increased with the number of features, stabilizing around
10 features, with the highest performance at 17 features (Fig. 2a). Therefore, we selected 17 explanatory variables to train the
ResNet, including five spectral bands, three vegetation indices, three topographic factors, three climate variables, geographic
location, tree cover, and land cover. Feature importance analysis showed that among the 17 explanatory variables, tree cover
contributed the most to the model, followed by slope and longitude. The near-infrared band and EVI also demonstrated high
importance, while other spectral features generally ranked lower than climate and topographic variables (Fig. 2b).

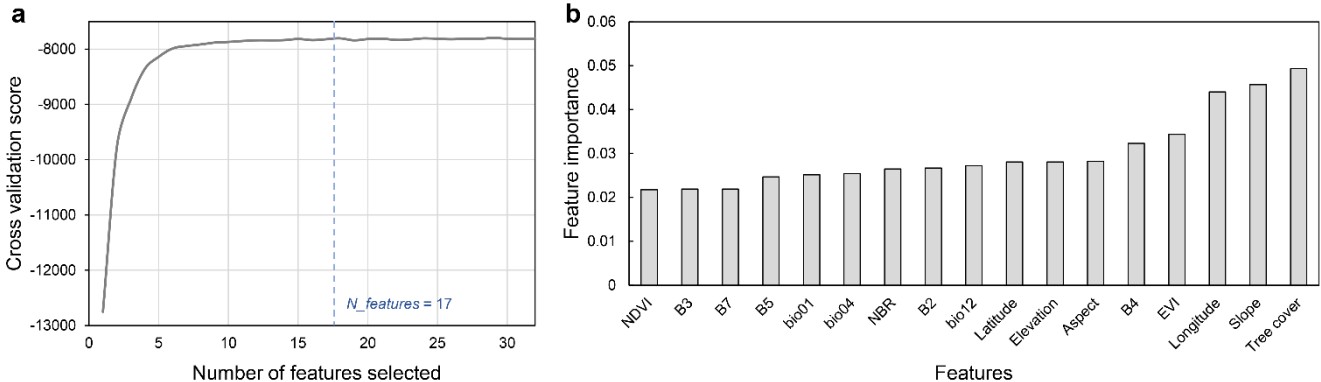

**Figure. 2:** Feature selection and importance analysis. (a) Cross-validation scores of the model with different numbers of features, measured
by negative mean squared error. (b) Feature importance distribution based on Random Forest.



### 3.2 Assessment evaluation

#### 3.2.1 Model performance

The ResNet model achieved high accuracy in both training (R² = 0.99, RMSE = 4.87 Mg/ha, Bias = -0.44 Mg/ha) and
testing (R² = 0.92, RMSE = 16.06 Mg/ha, Bias = -0.06 Mg/ha), demonstrating robust generalization capabilities without
indications of overfitting. Across all geographic regions, the model displayed consistent performance, with R² values ranging
from 0.87 to 0.96 and unbiased estimates (Bias range: -0.58 to 2.08; Table 1). The Central region achieved the highest testing
R² (0.96) and the lowest RMSE (10.10 Mg/ha), while the Three-North regions (Northeast, North, and Northwest) exhibited
slightly lower accuracy. In ecoregions, the model also performed well, maintaining unbiased estimates with testing R² > 0.81
and RMSE values ranging from 7.62 Mg/ha to 28.64 Mg/ha across seven ecoregions (Table 2). Although accuracy assessments
indicate that the model may exhibit higher RMSE in tropical rainforest regions with high AGBD, the ResNet model
outperformed other machine learning ensemble models (e.g., random forest, XGBoost, and LightGBM) in its ability to mitigate
the effects of spectral saturation (see Text S3 for details).

**Table 1.** Model performance on GEDI AGBD samples across different regions.

| Regions | Training | | | Testing | | |
|---|---|---|---|---|---|---|
| | $R^2$ | RMSE (Mg/ha) | Bias (Mg/ha) | $R^2$ | RMSE (Mg/ha) | Bias (Mg/ha) |
| Northeast | 0.99 | 5.55 | 0.59 | 0.89 | 19.30 | -0.58 |
| North | 0.99 | 3.44 | 0.13 | 0.87 | 14.47 | -0.21 |
| Northwest | 0.99 | 3.51 | 0.44 | 0.89 | 13.23 | -0.13 |
| East | 0.99 | 5.52 | -0.01 | 0.92 | 17.89 | 0.21 |
| Central | 0.99 | 5.00 | 0.37 | 0.96 | 10.10 | -0.08 |
| Southwest | 0.99 | 6.79 | 0.80 | 0.93 | 19.83 | 0.62 |
| South | 0.99 | 6.42 | 0.50 | 0.91 | 10.26 | 2.08 |
| The nation | 0.99 | 4.87 | -0.44 | 0.92 | 16.06 | -0.06 |


**Table 2.** Model performance on GEDI AGBD samples across different ecoregions.

| Ecoregions | Training | | | Testing | | |
|---|---|---|---|---|---|---|
| | $R^2$ | RMSE (Mg/ha) | Bias (Mg/ha) | $R^2$ | RMSE (Mg/ha) | Bias (Mg/ha) |
| Temperate desert | 0.99 | 3.27 | 0.38 | 0.81 | 13.10 | -0.04 |
| Tibetan plateau alpine vegetation | 0.97 | 4.72 | 0.61 | 0.92 | 7.62 | 0.69 |



| | | | | | | |
|---|---|---|---|---|---|---|
| Subtropical evergreen broadleaf forest | 0.99 | 6.80 | 0.63 | 0.88 | 23.00 | 0.31 |
| Tropical monsoon rainforest and rainforest | 0.99 | 10.33 | 0.27 | 0.93 | 28.64 | 1.41 |
| Warm temperate deciduous broadleaf forest | 0.99 | 4.39 | 0.12 | 0.83 | 14.53 | -0.77 |
| Temperate coniferous and deciduous broadleaf mixed forest | 0.99 | 6.10 | 1.21 | 0.85 | 27.74 | -1.15 |
| Cold temperate coniferous forest | 0.99 | 3.27 | -1.68 | 0.95 | 18.26 | -0.26 |

### 3.2.2 Independent accuracy evaluation

The validation using historical field survey data confirms that the AGBD estimates exhibit reasonable accuracy across different time periods. For the period 1985–2008, the model achieved an R² of 0.63, an RMSE of 68.26 Mg/ha, and a bias of -19.87 Mg/ha (Table 3, Fig. 3). Among ecoregions, the cold temperate coniferous forest showed the highest accuracy, with the lowest RMSE (25.73 Mg/ha). Similarly, the temperate coniferous and deciduous broadleaf mixed forest achieved low RMSE (30.74 Mg/ha) and minimal bias (4.36 Mg/ha). In contrast, the tropical monsoon rainforest and rainforest regions presented greater challenges, with higher RMSE (90.82 Mg/ha) and a significant negative bias (-68.57 Mg/ha). Other ecoregions, such as the temperate desert and subtropical evergreen broadleaf forest, displayed intermediate accuracy levels, with RMSEs of 47.82 Mg/ha and 67.83 Mg/ha and biases of 29.54 Mg/ha and -36.48 Mg/ha, respectively.

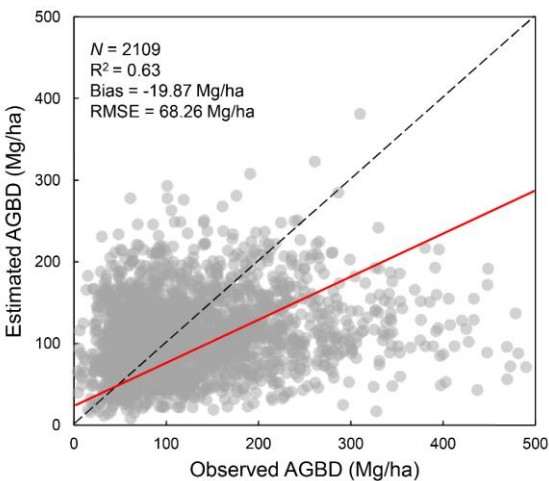

**Figure 3:** Scatter plot comparing field survey data with ResNet-estimated AGBD. The black dashed line represents the 1:1 line, while the red solid line represents the fitted regression line.

**Table 3.** Independent accuracy evaluation using field survey data across different ecoregions.

| Ecoregions | $n$ | RMSE (Mg/ha) | Bias (Mg/ha) |
|---|---|---|---|
| Temperate desert | 4 | 47.82 | 29.54 |
| Subtropical evergreen broadleaf forest | 1629 | 67.83 | -36.48 |
| Tropical monsoon rainforest and rainforest | 56 | 90.82 | -68.57 |
| Warm temperate deciduous broadleaf forest | 314 | 54.29 | 35.09 |
| Temperate coniferous and deciduous broadleaf mixed forest | 66 | 30.74 | 4.36 |
| Cold temperate coniferous forest | 39 | 25.73 | 12.11 |
| The nation | 2109 | 68.26 | -19.87 |


The correlation analysis between provincial-level AGC estimates and NFI forest AGC statistics reveals strong correlations, with coefficients consistently exceeding 0.85 ($p < 0.05$) across different assessment periods, ranging from 0.85 to 0.89 (Fig. 4). These results suggest that the AGC estimates effectively capture the temporal changes in national AGC, as reflected in the NFI's multi-period data. The consistent correlations observed across the various assessment periods

demonstrate that the AGC estimates are reliable indicators of forest carbon stock dynamics over time.

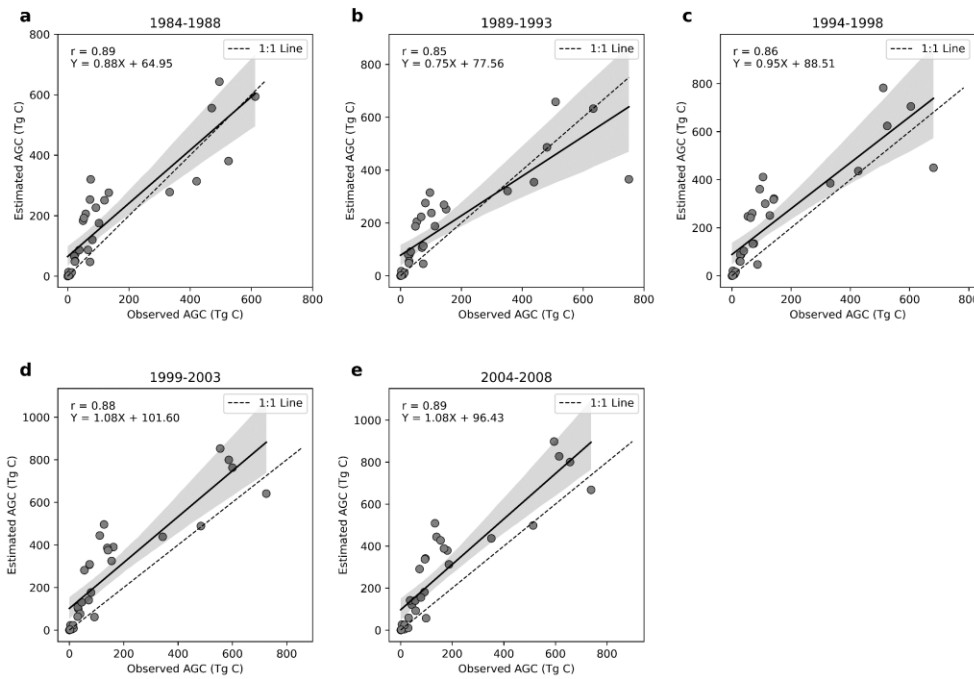

**Figure 4:** Scatter plot comparing field survey data with ResNet-estimated AGBD. The black dashed line represents the 1:1 line, while the red solid line represents the fitted regression line.



### 3.2.3 Residual analysis

The residual analysis based on geographic location, tree cover, and terrain underscores the robustness of the AGBD estimates across various conditions (Fig. 5). We found that the residual range is wider in low latitude (<25°N) and high latitude (>40°N) regions compared to mid-latitude areas. As slope and tree cover increase, the model's residuals also broaden, indicating that the predictions may exhibit larger errors in regions with low or high latitudes, steep slopes, and dense forests. This trend likely reflects the influence of tree structure and terrain complexity on forest biomass estimation. No significant

biases were observed across different longitudes, aspects, or elevations (Fig. 5), indicating small differences between predicted and observed values. This suggests that AGBD predictions remain relatively consistent and unaffected by these factors, with minimal RMSE across varying terrain features. In summary, the results demonstrate that the comprehensive technical framework provided in this study effectively meets the GCOS core principle that AGBD estimates should be as unbiased as possible (GCOS, 2017).

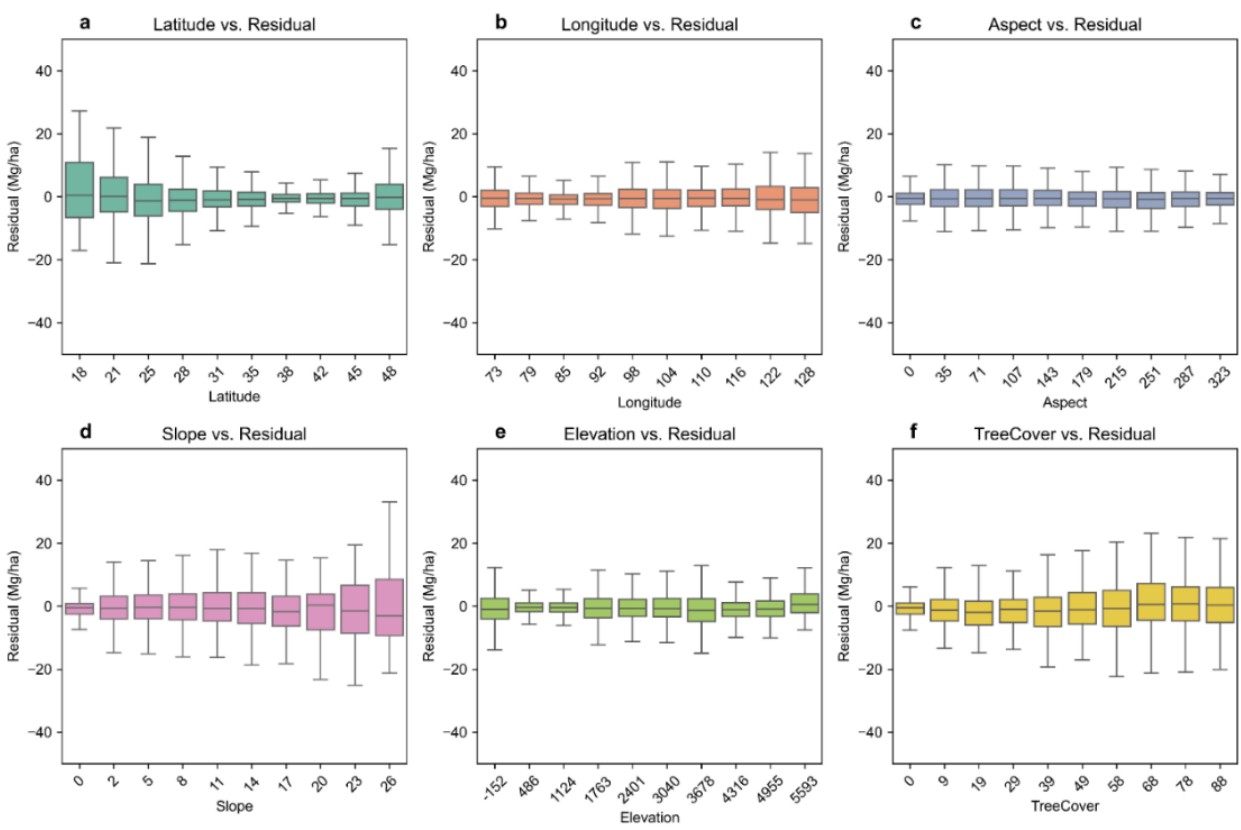


**Figure 5:** Residual analysis for forest AGBD estimation. Panels (a)–(f) depict AGBD prediction residuals under varying latitudes (°N), longitudes (°E), aspects (°), slopes (°), elevations (m), and tree cover (%), respectively. Positive residuals indicate predicted values higher than reference values, while negative residuals signify underestimation of AGBD compared to GEDI references. The box plot displays the median, quartiles, as well as the 10th and 90th percentiles of the residuals. The

analysis was performed based on the 20% of 2019–2021 GEDI AGBD footprints and their corresponding estimates.



### 3.3 Spatial pattern of forest AGB

The 30m resolution mapping of China's forest AGBD reveals substantial spatial variability in biomass density (Fig. 6a). In 2023, the national average forest AGBD was 122.69 Mg/ha (Fig. 6b), with a total forest AGB stock amounting to 13.97 PgC (Fig. 6c). Among the seven geographic regions, the Southwest (142 Mg/ha), Northeast (123 Mg/ha), and Northwest (123 Mg/ha) regions exhibited biomass densities above the national average. High-density areas with forest AGBD exceeding 300 Mg/ha were predominantly located in these regions, including southern Tibet, the Qinling Mountains, and parts of Northeast China. Additionally, central Taiwan exhibited several hotspots with forest AGBD greater than 300 Mg/ha. These regions are notably rich in forest resources and represent significant areas of natural forest distribution in China. Conversely, the northern, central, eastern, and southern regions have historically been subjected to frequent human disturbances and high-intensity impacts, leading to younger forest stands and relatively lower biomass densities. Nonetheless, these regions present a substantial potential for carbon sequestration. The provinces of Yunnan, Sichuan, Heilongjiang, Guangxi, and Tibet are the top five in terms of forest biomass carbon stocks, collectively accounting for 41.5% of the national total (Fig. 6c). This highlights the uneven spatial distribution of forest resources across China.

In 2023, the tropical rainforest exhibited the highest AGBD (165.17 Mg/ha), followed by the temperate coniferous and deciduous broadleaf mixed forest (136.38 Mg/ha), subtropical evergreen broadleaf forest (123.88 Mg/ha), warm temperate deciduous broadleaf forest (111.08 Mg/ha), cold temperate coniferous forest (107.77 Mg/ha), temperate desert region (97.50 Mg/ha), and Tibetan plateau alpine region (72.98 Mg/ha).

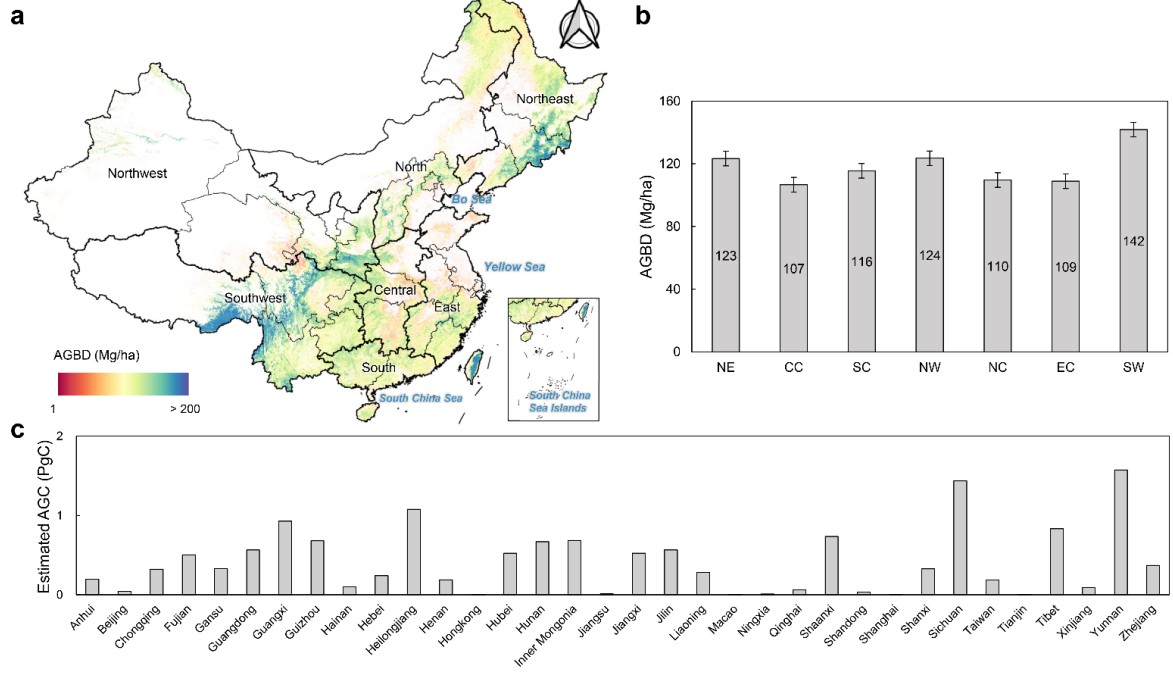

**Figure 6:** Spatial pattern of forest AGB and regional overview. (a) The spatial pattern of forest AGBD in China in 2023. (b) The average of AGBD across difference geographical regions. (c) The estimated forest AGC stock of difference provinces.





The pixel-level analysis reveals that the average uncertainty in AGBD estimates across the study area is 13.94 Mg/ha, with individual pixel uncertainties ranging from 0 Mg/ha to 85 Mg/ha (Fig. 7a). The vast majority of the study area (99.35%) exhibits an uncertainty of less than 30 Mg/ha, while only a small fraction (0.65%) has uncertainties of 30 Mg/ha or greater (Fig. 7b). Notably, uncertainties are higher than the national average in the southwestern and southern regions, at 15.55 Mg/ha

and 16.62 Mg/ha, respectively. In contrast, the northwestern and northern regions have relatively lower uncertainties, with estimates falling below 10 Mg/ha (Fig. 7c). Overall, the uncertainty in forest AGBD across the study area is well-controlled, indicating a reliable dataset. However, the higher uncertainties in specific regions underscore the need for further refinement in these areas, particularly in complex terrains where topographic and vegetation heterogeneity may challenge the accuracy of AGBD estimates (Fig. 5d).

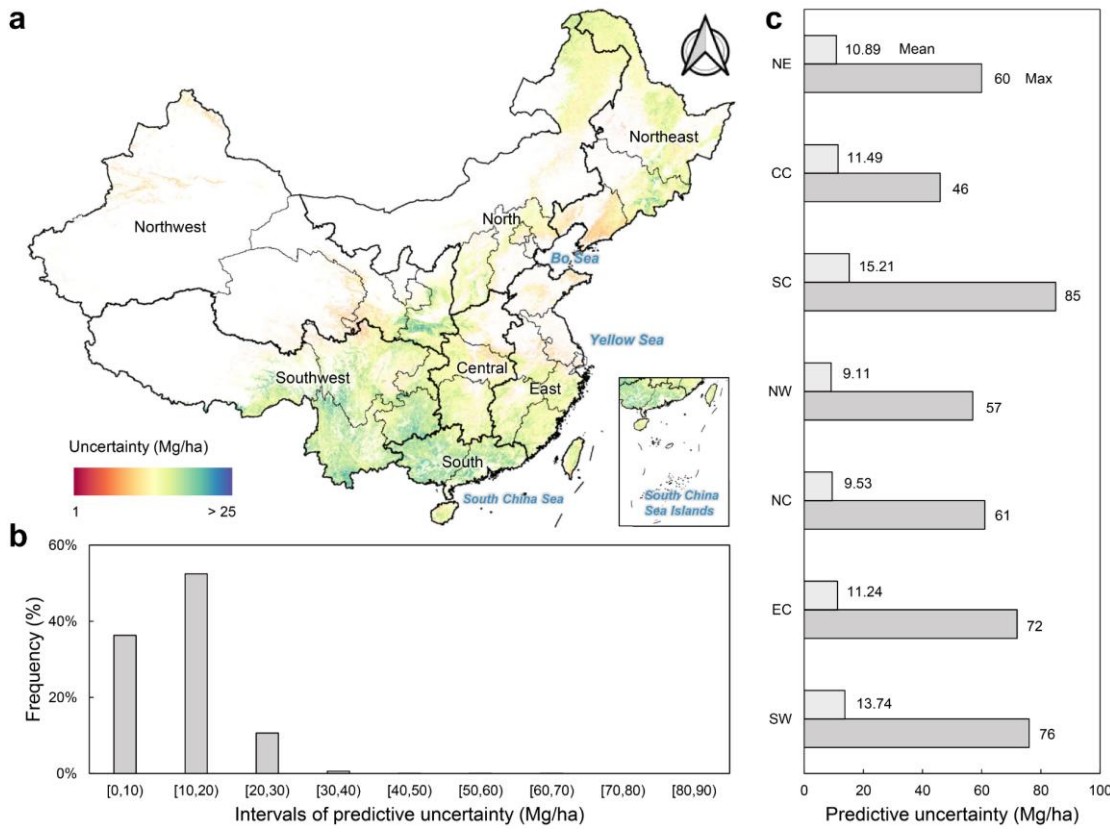


**Figure 7:** Uncertainty analysis of forest AGB in China for the year 2023. (a) Predictive uncertainty map (expressed as standard deviation) of forest AGBD across China. (b) Frequency distribution of predictive uncertainty, categorized into 10 Mg/ha intervals. (c) Regional statistics of predictive uncertainty across different geographical areas.

**3.4 Temporal trend and dynamics of forest AGBD in China**

380       The temporal analysis of China's average forest AGBD reveals a significant upward trend over the study period, which the average AGBD increased from 95.74±11.30 Mg/ha in 1985 to 122.69±13.94 Mg/ha in 2023 (Fig. 8b). The average AGBD



grew at an annual rate of 0.69 Mg ha⁻¹ yr⁻¹, representing a 28.1% increase over the entire period. Most regions experienced an increase in AGBD during the study period, with the exception of certain areas in northeastern and northern China, where agricultural expansion and deforestation likely contributed to forest AGBD reduction (Fig. 8a). In contrast, significant

increases in AGBD were observed in the Three-North Shelterbelt region, as well as in southwestern and southern China. In the Three-North region, extensive afforestation and the construction of shelter forests transformed non-forested areas into forested land, resulting in a substantial increase in AGBD. Similarly, large-scale afforestation efforts as part of desertification control in the southwestern and southern (e.g., Guangxi) regions, also contributed to a significant rise in AGBD. Among the seven geographical regions, the southern region experienced the fastest growth in AGBD (0.90 Mg ha⁻¹ yr⁻¹), followed by the

northeast (0.81 Mg ha⁻¹ yr⁻¹), southwest (0.79 Mg ha⁻¹ yr⁻¹), north (0.73 Mg ha⁻¹ yr⁻¹), northwest (0.67 Mg ha⁻¹ yr⁻¹), central (0.41 Mg ha⁻¹ yr⁻¹), and eastern regions, which exhibited the slowest growth rate (0.26 Mg ha⁻¹ yr⁻¹) (Fig. 8b).

We observed increasing AGBD trends across most ecoregions, except for the Tibetan plateau alpine region, where forest AGBD declined from 80.07 Mg/ha in 1985 to 72.98 Mg/ha in 2023, with a trend of -0.12 Mg ha⁻¹ yr⁻¹. Alpine forests, being more sensitive to environmental changes, may have experienced degradation due to climate warming and extreme weather

events. The tropical rainforest showed the most pronounced increase in AGBD (1.17 Mg ha⁻¹ yr⁻¹), followed by the temperate coniferous and deciduous broadleaf mixed forest (1.00 Mg ha⁻¹ yr⁻¹), subtropical evergreen broadleaf forest (0.67 Mg ha⁻¹ yr⁻¹), cold temperate coniferous forest (0.64 Mg ha⁻¹ yr⁻¹), temperate desert region (0.60 Mg ha⁻¹ yr⁻¹), and warm temperate deciduous broadleaf forest (0.48 Mg ha⁻¹ yr⁻¹).

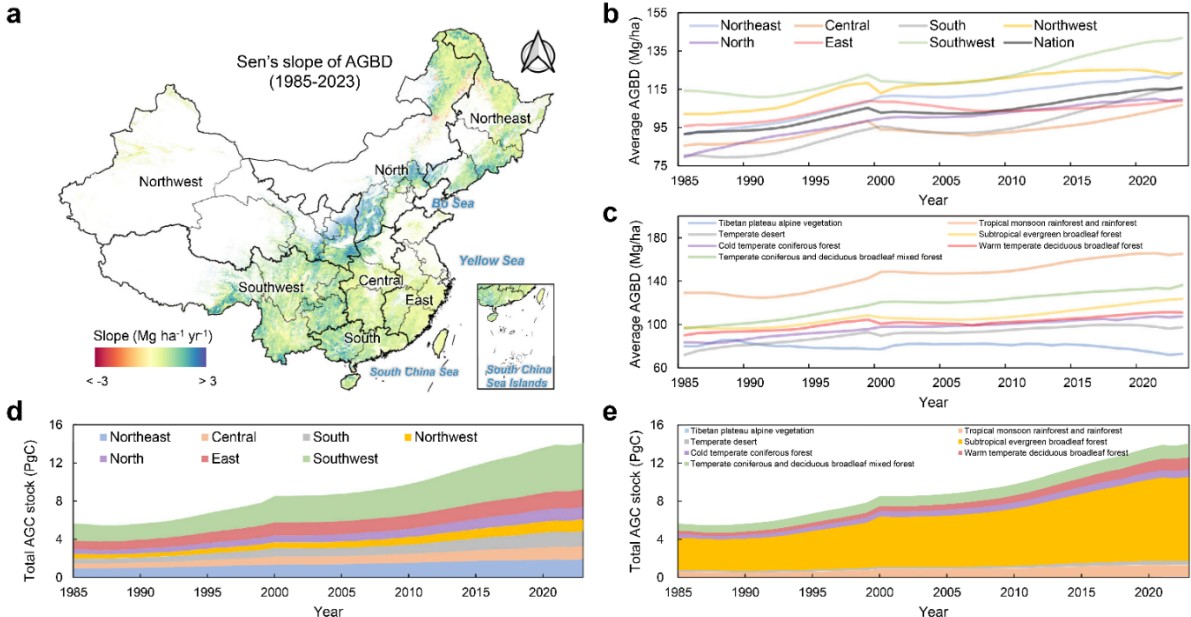

**Figure 8:** Long-term (1985–2023) trends in forest AGB and AGC stocks in China. (a) Pixel-level forest AGBD trend map derived from Sen's slope analysis (Sen, 1968). (b) and (c) display annual changes in forest AGBD from 1985 to 2023 across geographical regions and ecoregions, respectively. (d) and (e) illustrate annual changes in total forest AGC stock over the same period, also across geographical regions and ecoregions. AGC is calculated from biomass using a conversion factor of 0.5.



### 3.5 China's Forests as a strong carbon sink over the past four decades

The spatial distribution of China's forest aboveground carbon (AGC) stock in 2023 was uneven. The southwestern region held the largest share, with 4.84 PgC, accounting for over one-third of the national total (34.43%). The northeast contributed 13.65%, followed by the east (12.94%), the south (11.32%), the central region (9.80%), the north (9.19%), and the northwest (8.66%) (Fig. 8d). Disparities in AGC stock were even more pronounced across different forest types. The subtropical evergreen broadleaf forest contained 8.82 PgC, making up over half of the national total (62.80%), followed by the temperate

coniferous and deciduous broadleaf mixed forest (10.32%), warm temperate deciduous broadleaf forest (9.11%), tropical rainforest (8.90%), cold temperate coniferous forest (5.44%), temperate desert region (2.91%), and the Tibetan Plateau alpine region (0.55%) (Fig. 8e).

         Throughout the study period, China's forest AGC showed an upward trend, with total forest AGC stock expanding from 5.50±0.23 PgC in 1985 to 13.97±0.87 PgC in 2023 (Fig. 8d, e). Across all geographic and ecological regions, total forest AGC

stock consistently increased from 1985 to 2023. The southwestern region experienced the fastest increase (0.09 PgC per year), while growth rates in other regions were relatively similar (0.02 PgC per year). Among ecoregions, the subtropical evergreen broadleaf forest showed the fastest growth (0.15 PgC per year), while the Tibetan Plateau alpine region exhibited the slowest growth (0.001 PgC per year). However, net carbon storage initially declined at a rate of -0.02 PgC per year during 1985–1989. This trend reversed after 1990, with a robust increase of 0.25 PgC per year. Overall, the total forest AGC stock increased at a

rate of 0.22±0.01 PgC per year, effectively doubling China's forest carbon reserves over the past 40 years, achieving a remarkable growth of 154.0% (see Text S4 for discussion). The forest net AGC sink offset 11.5%–14.9% of fossil fuel and industrial emissions between 1985 and 2023.

### 3.6 Forest growth dominated AGC stock increases in China

         This study highlights the dual role forests played in carbon sequestration and emissions in China during the period from

1985 to 2023, depending on the nature of forest changes. Forest growth, which involves increased tree cover in already forested areas, was a major contributor to carbon sequestration, particularly in regions south of the Qinling Mountains (Fig. 9a). The spatial changes in AGC induced by forest growth closely mirror the age distribution of China's forests. The carbon sequestration per unit area due to forest growth was significantly higher in southern regions compared to northern, northeastern, southwestern, and Taiwan, where natural forests dominate. These older forests with higher canopy cover had limited potential

for further AGC increase, following the "law of constant final yield" (Adler et al., 2018), which explains why AGC growth in younger and less dense forests in central, southern, and eastern China outpaced that of other regions, reflecting the effects of long-term conservation efforts.

         Forest expansion also contributed to additional carbon sequestration by converting non-forested land to forest, with significant gains observed in the southwest and western regions (Fig. 9b). This is closely related to reforestation efforts initiated

since 1985. These areas, previously allocated for other land uses, significantly boosted carbon stocks through afforestation




Earth System
Science
Data

initiatives. Notably, we observed that carbon sequestration per unit area from forest expansion was considerably lower than that from forest growth, which can be explained by the fact that AGC accumulation rates follow a "slow-fast-slow" pattern as forests mature (see Fig. 11 for illustration). Hence, the carbon sequestration in newly planted forests was naturally lower than that in pre-existing forests. However, the carbon sequestration due to forest expansion in the southwest was significantly higher

than in other regions, aligning with recent studies showing that the carbon sink potential in southwest China may have been historically underestimated (Wang et al., 2020). Conversely, tree cover loss and forest-to-non-forest transitions were the primary contributors to carbon emissions, mainly concentrated in northern and northeastern regions (Fig. 9c, d). The spatial patterns of carbon emissions due to forest loss suggest that deforestation and land-use conversion were concentrated in the same regions experiencing high developmental pressure and agricultural expansion.

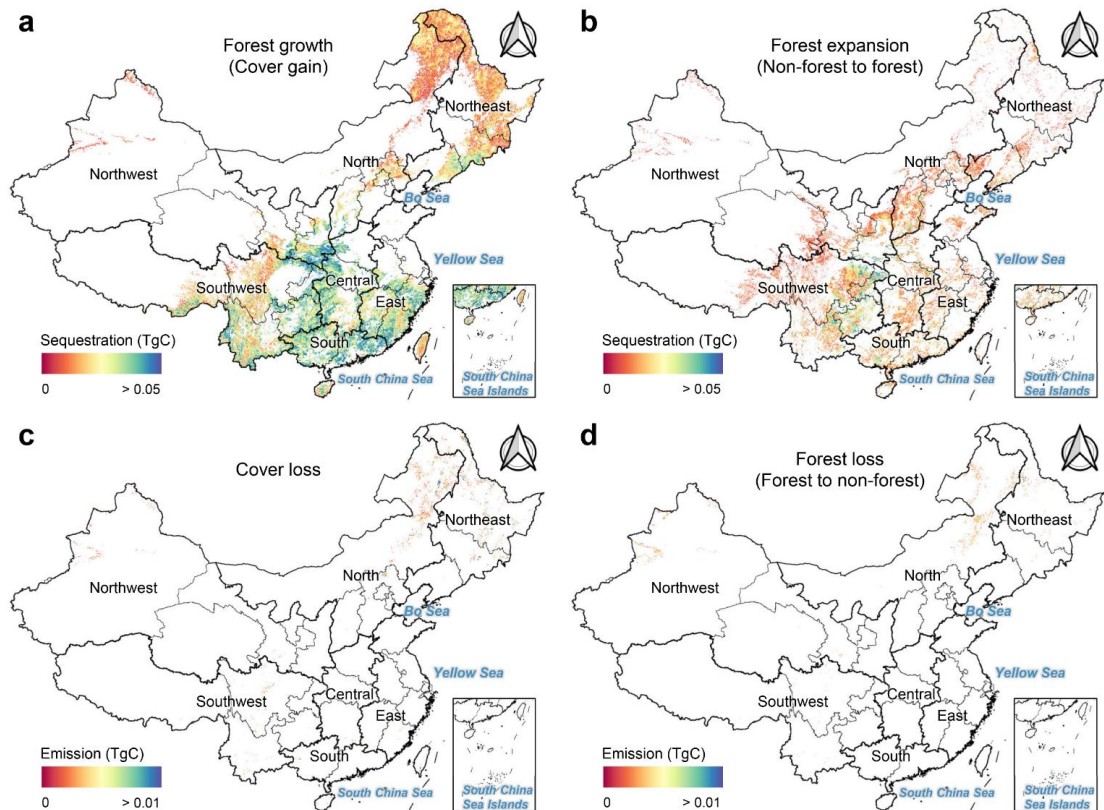

**Figure 9:** AGC Spatial changes induced by forest dynamics in China between 1985 and 2023. (a) and (b) represent the spatial distribution of carbon sequestration driven by forest growth and expansion, respectively. (c) and (d) illustrate the spatial distribution of carbon emissions resulting from tree cover loss and forest loss, respectively.

We observed that approximately 65.1% (5.75 PgC) of the carbon sequestration was due to forest growth, while the remaining 34.9% (3.09 PgC) resulted from forest expansion (Fig. 10). Although forest growth contributed more to the AGC stock increment, this finding suggests that afforestation activities since 1985 have significantly boosted AGC, accounting for nearly one-third of the total increase. These initiatives have involved planting new forests on previously non-forested land and



restoring degraded forest areas, thus expanding China's forested landscapes and bolstering their carbon sequestration capacity.
This substantial contribution underscores the positive impact of forest conservation policies and enhanced management practices, which have facilitated forest recovery, maturation, and increased carbon sequestration over time. In terms of carbon emissions, tree cover loss accounted for 47.77% of the AGC reduction, while forest loss (forest to non-forest transitions) was responsible for 52.23%. This indicates that complete deforestation had a more substantial impact on AGC loss than reductions in canopy density.

The contrasting spatial dynamics between carbon sequestration and emissions underscore the complexity of AGC changes over the past four decades. While large areas of China have acted as carbon sinks due to reforestation and forest regeneration, other regions have simultaneously contributed to significant carbon emissions through deforestation and land degradation. This dual impact highlights the critical importance of targeted forest management strategies aimed at enhancing carbon sequestration while minimizing deforestation and forest degradation, ensuring a balanced and sustainable approach to forest
carbon dynamics in the future.

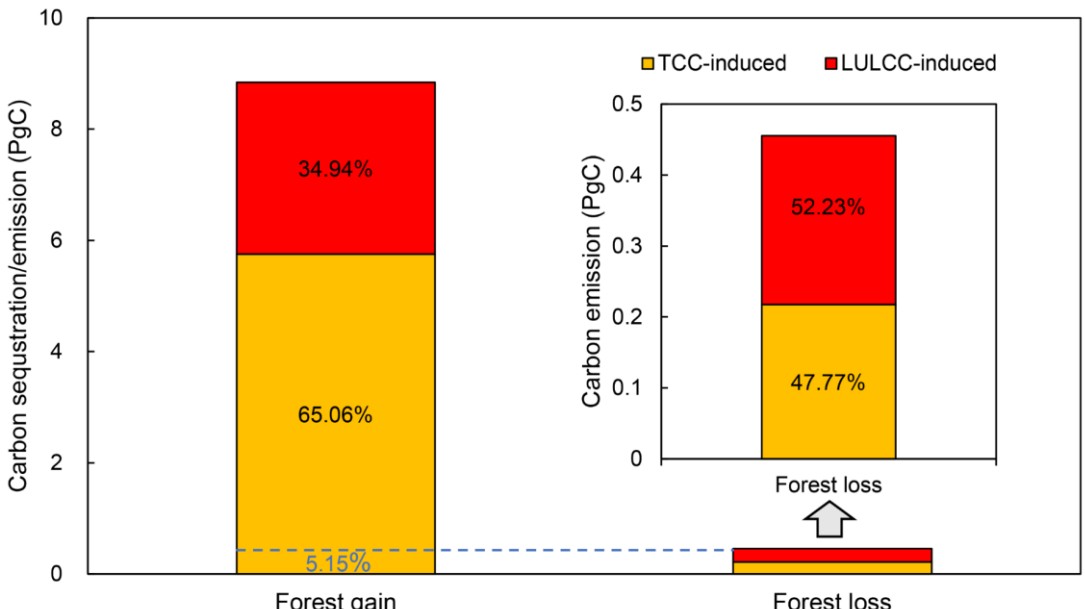

**Figure 10:** Quantification of AGC changes induced by forest dynamics in China between 1985 and 2023.

## 4 Discussion

### 4.1 Insights into the CFATD

The typical examples of forest growth and natural forest degradation detected by CFATD highlight its effectiveness in capturing changes in forest AGB (Figs. 11–12). In the case illustrated in Fig. 11, CFATD identifies the establishment of shelterbelt forests in the Three-North regions, showing a consistent increase in AGBD since 1994. This trend follows the



sigmoidal growth curve typical of young forests, characterized by phases of growth transitioning from "slow" to "accelerating" and then to "stabilizing." In the first growth phase (1994–1999), AGBD increased at a rate of 5.60 Mg ha⁻¹ yr⁻¹, which

accelerated to 8.45 Mg ha⁻¹ yr⁻¹ during the second phase (2000–2007) before slowing to -0.03 Mg ha⁻¹ yr⁻¹ in the final phase (2008–2023).

Another case focuses on the degradation of natural dark coniferous forests in the alpine valleys of Sichuan and Yunnan (Fig. 12). Despite stable forest cover and even an expansion in forest area, CFATD detects a clear decline in AGBD. AGBD fell from 292 Mg/ha in 1985 to 136 Mg/ha in 2023, with a degradation rate of -3.15 Mg ha⁻¹ yr⁻¹. This indicates that the forest

has experienced continuous degradation since the data became available, with no signs of recovery.

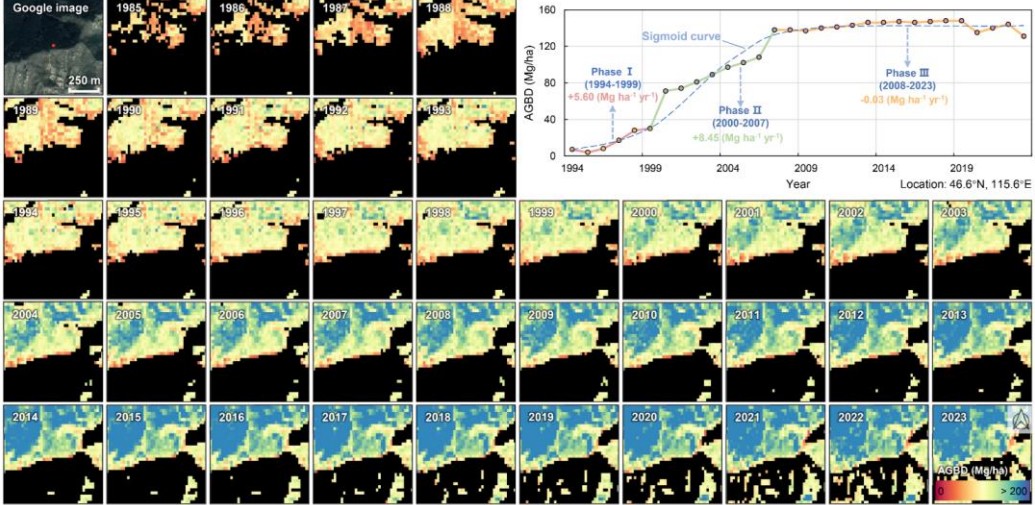

**Figure 11:** Increase in AGBD of newly planted forests under Three-North Shelter Forest Program from 1985 to 2023, as depicted by the CFATD (Satellite imagery: © Google Earth).

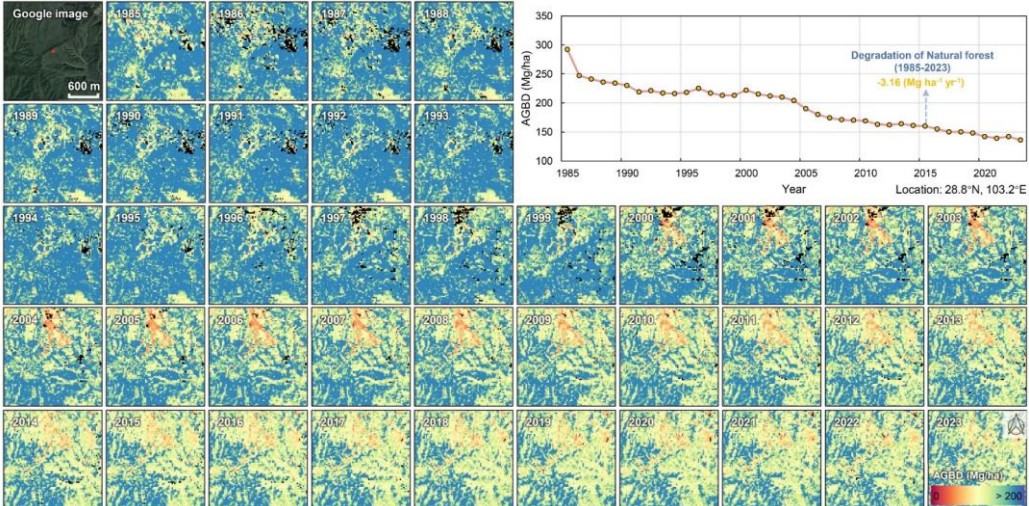

**Figure 12:** Decline in AGBD of degraded natural forests in southwest China from 1985 to 2023, as depicted by the CFATD (Satellite imagery: © Google Earth).



## 4.2 Comparison with external AGBD datasets

### 4.2.1 Static and spatial comparison

In pixel-level comparisons of our AGBD map against datasets from Zarin et al., (2016), Hu et al., (2016), Su et al., (2016),
Santoro et al., (2021), Yang et al., (2023), and ESA CCI (see Text S2 for more information about the external AGBD datasets),
the average differences were -5.86 Mg/ha, -45.23 Mg/ha, 24.78 Mg/ha, 47.91 Mg/ha, 19.55 Mg/ha, and 60.14 Mg/ha,
respectively (Fig. 13). Our map aligns more closely with Zarin et al., Su et al., Yang et al., while larger discrepancies are
observed with the Hu, Santoro, and ESA CCI datasets. Specifically, the percentage of pixels with absolute differences less
than 50 Mg/ha and 25 Mg/ha, respectively, are as follows: Zarin et al. (49.5%, 25.9%), Hu et al. (50.7%, 27.6%), Su et al.
(59.8%, 33.9%), Santoro et al. (53.2%, 26.4%), Yang et al. (65.8%, 39.0%), and ESA CCI (38.1%, 18.2%).

In southern, eastern, and southwestern China, our estimates are over 50 Mg/ha lower than Zarin, Hu, and Su, whose
coarse spatial resolution likely overestimates AGBD in fragmented forests. Temporal mismatches and intensive human
activities in these regions exacerbate the differences. Conversely, our estimates in northern and northwestern regions are higher
than Zarin and Su but align well with Hu, with minimal differences (<25 Mg/ha). Compared to Santoro and ESA CCI, our
map consistently reports higher AGBD across China, as their global focus and lack of region-specific samples likely lead to
underestimations. The Santoro and ESA CCI maps, which report average AGBD of 57.05 Mg/ha and 64.88 Mg/ha for 2010
and 2020, respectively, represent the lowest levels listed in Table S1. The Yang map shows strong spatial consistency with
ours but underestimates AGBD in hotspots like the Qinling Mountains, southern Tibet, and Taiwan province due to limited
field samples and spectral saturation.



**Figure 13:** Difference between the generated AGBD map and external AGBD datasets. (a)–(f) are pixel-level differences between the estimated forest AGBD from CFATD and the forest AGBD from Zarin (Zarin et al., 2016), Hu (Hu et al., 2016), Su (Su et al., 2016), Yang (Yang et al., 2023), and ESA CCI, respectively (the estimated AGBD minus external AGBD).

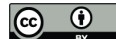

Key differences among datasets stem from their capacity to capture spatial variability, manage spatial resolution, and
comprehensively represent forested regions. Our map captures fine-scale patterns, such as higher AGBD in forest interiors and
lower AGBD at edges (Fig. 14), outperforming Yang in variability representation, while Zarin, Santoro, and ESA CCI also
perform well in this aspect. Coarse-resolution datasets, such as the Hu map (Fig. 14f) and Su map (Fig. 14g) (1,000 m), show
significantly reduced accuracy in fragmented landscapes, making them unsuitable for precise carbon accounting. Furthermore,
the quality of forest thematic maps plays a critical role in AGBD mapping; only CFATD, Zarin, and Santoro effectively
detected shelterbelts within agricultural areas, whereas other datasets either partially missed or entirely failed to capture these
features (The third scenario of Fig. 14).

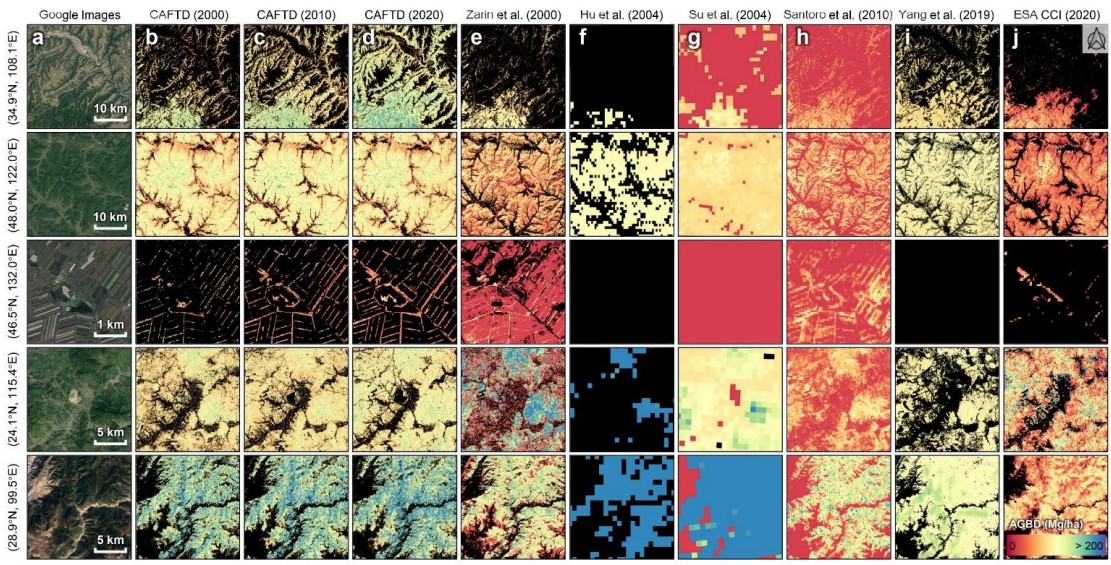

**Figure 14:** Comparison of AGBD product details across different datasets. (a) Google high-resolution imagery (© Google
Earth); (b)–(d) represent the CFATD products for the years 2000, 2010, and 2020, respectively; (e)–(j) display AGBD maps
from Zarin et al., Hu et al., Su et al., Santoro et al., Yang et al., and ESA CCI. Each row corresponds to subregions representing
different forest types. Note: The numbers in parentheses indicate the corresponding year of each dataset.

**4.2.2 Dynamic and trend comparison**

Our AGBD dataset reveals a consistent upward trend from 1985 to 2023, highlighting the steady recovery and effective
management of China's forests (Fig. 15). This trend closely aligns with national policies on forest conservation, afforestation,
and reforestation, offering a robust depiction of changes in forest carbon stocks over time (Zeng et al., 2023). Four time-series
AGB datasets were used for comparison (see Text S2 for more information about the external AGB datasets for dynamic and
trend comparison). The Hengeveld dataset (Hengeveld et al., 2015) shows fluctuations, including declines in AGBD during
the late 1990s and early 2000s. Similarly, the ESA CCI dataset, though global in scope, indicates a general decline in AGBD,
particularly during the 2010s. These discrepancies can be attributed to the coarse spatial resolution and global-scale models
used in those studies, which may not capture localized forest dynamics in China, especially in areas experiencing rapid land-
use changes.

Our findings are consistent with previous studies, including those by Liu et al., (2015) and Chen et al., (2023), both of which also report a long-term upward trend in China's forest AGBD. Furthermore, our dataset aligns closely with estimates derived from forest inventory data (87.06–89.82 Mg/ha for the period 1984–1998, Fang et al., 2001) and remote sensing data

(90.62 Mg/ha in average for the period 1981–1999, Piao et al., 2005). In contrast, Chen's dataset reports a lower overall AGBD, likely due to methodological differences in spatial resolution, as well as forest area definitions and scopes. Chen's study applies a forest definition based on >15% tree cover, which results in mixed pixels containing other vegetation types (e.g., shrubs, herbs, and crops), leading to an underestimation of average forest AGBD in China. However, when aggregated with the forest cover map used by Hu et al., (2016) at 1 km resolution, the average AGBD in Chen's map increases, ranging from 68.94 Mg/ha

in 2002 to 75.58 Mg/ha in 2021.

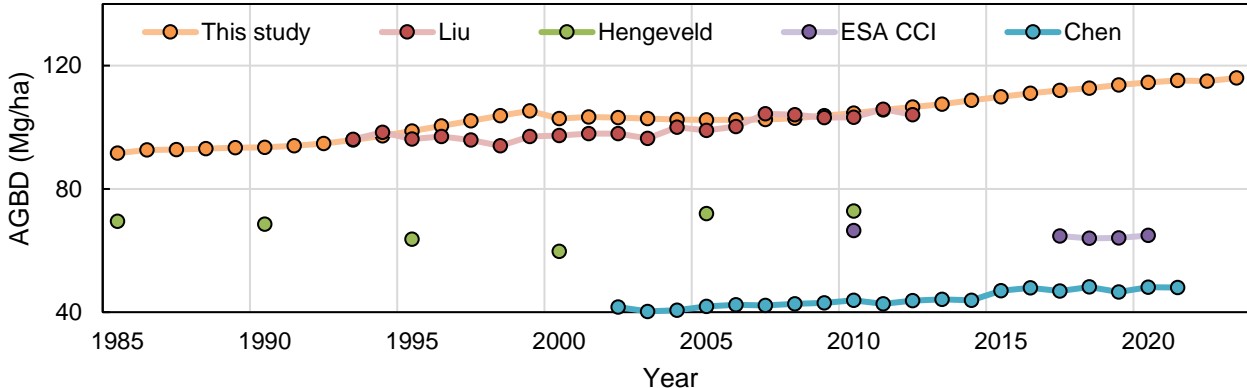

**Figure 15:** Comparative analysis of temporal dynamics across different AGB products.

### 4.3 Influence of forest definition on forest AGB estimation

Forests are defined according to diverse criteria that vary across institutional and scientific contexts. Among these, tree

cover thresholds of 10%, 20%, and 30% are the most commonly used by parties to the United Nations Framework Convention on Climate Change (UNFCCC). For example, the FAO defines forests using a canopy cover threshold of ≥10%, a minimum tree height of ≥5 meters, and a minimum area of 0.5 hectares (FAO, 2020). In contrast, regional or national inventories often adopt stricter or more lenient thresholds to better suit local ecological conditions. These discrepancies lead to variations in forest extent, which influence the forest AGB and its carbon sink (see Text S4) estimates derived from these definitions. A

broader forest definition, such as one with a lower canopy cover threshold, may include areas with sparse vegetation, resulting in higher total AGB estimates but lower average AGBD. Conversely, stricter definitions that focus on dense, high-biomass forests yield higher AGBD estimates but lower total AGB. A previous study demonstrated that defining forests with a 10% tree cover threshold increases the total forest area by 4.3% and 6.2% compared to thresholds of 25% and 30%, respectively (Su et al., 2016). Such differences in forest area can lead to a 3.2%–4.9% variation in China's total AGB estimates for 2004,

equivalent to approximately 0.56–0.85 Pg of forest biomass.





### 4.4 Limitations and future improvement

Current AGBD maps for China's forests often overestimate low AGBD and underestimate high AGBD. The persistent pattern of overestimating low biomass density and underestimating high biomass density in spatially derived AGBD maps is well-documented (Réjou-Méchain et al., 2019; Rodríguez-Veiga et al., 2019). These discrepancies are largely driven by sensor
limitations, such as the saturation of optical and radar sensors at higher AGBD values (Rodríguez-Veiga et al., 2019), leading to underestimation in dense forests. The overestimation of low AGBD is more complex and often arises from regression models that fit Earth observation data to limited plot data, especially when the plots are not fully representative of global forest conditions. Systematic biases in AGBD estimation can stem from various sources, including tree size measurement techniques, human and equipment errors, incorrect allometric models, and environmental factors such as precipitation and snow, which
can influence Earth observation signals (Santoro et al., 2021). In fact, we found that positional errors and mismatches between the pixel scale and plot size can easily distort AGBD estimates derived from remote sensing data. This highlights the need for improved plot matching techniques and more accurate field measurements to reduce uncertainties in future AGBD mapping efforts.

It is also important to note that the GEDI footprint AGBD dataset lacks ground AGBD data from China and other
neighbouring countries, such as Russia, during the model training process (Duncanson et al., 2022). Although the Level 4A (L4A) algorithm is designed to address geographic transferability, allowing predictions beyond the geographic range of the training data (Kellner et al., 2023), the absence of ground AGBD data from Asia in the development of the L4A product may still introduce uncertainty in AGBD estimations for China and the broader Asian region. Future studies should conduct more extensive evaluations of the accuracy of the GEDI AGBD product in China and Asia.
While deep learning has been employed to reduce the impact of spectral saturation and improve estimation accuracy, there are still key areas for future improvement in the accuracy and reliability of forest AGBD estimation. Integrating multi-source remote sensing data—such as optical, radar, and LiDAR—can address the limitations of individual sensors and mitigate saturation effects by providing a more comprehensive view of forest structure (see Text S5). Additionally, developing allometric models tailored to specific forest types and regions can enhance the precision of AGBD estimates, particularly in
heterogeneous landscapes. Enhancing calibration and validation efforts by increasing the number and diversity of ground-based measurements, especially in underrepresented regions, will also help reduce estimation biases. Incorporating time-series data into AGBD models can better capture temporal dynamics and distinguish between short-term fluctuations and long-term trends (Araza et al., 2022; Duncanson et al., 2019). Finally, advancing machine learning and deep learning techniques offers promising avenues for more accurate AGBD estimation. These methods can uncover complex relationships between remote
sensing data and forest biomass, addressing challenges like mixed-pixel effects and seasonal variations, thereby improving the overall quality of AGBD maps.





## 5 Data availability

The China Forest AGB Time Series Dataset (CFATD) generated by this study is accessible via the Google Earth Engine script at this link: https://code.earthengine.google.com/4f8ad8d32ddb84e826e941a95f31f9be. Users also can download the
CFATD from Zenodo: (Part I: 1985–1993) https://doi.org/10.5281/zenodo.12620984 (Cai et al., 2025a), (Part II: 1994–2001) https://doi.org/10.5281/zenodo.12637101 (Cai et al., 2025b), (Part III: 2002–2008) https://doi.org/10.5281/zenodo.12655492 (Cai et al., 2025c), (Part IV: 2009–2015) https://doi.org/10.5281/zenodo.12658255 (Cai et al., 2025d), (Part V: 2016–2021) https://doi.org/10.5281/zenodo.12742210 (Cai et al., 2025e), (Part VI: 2022–2023) https://doi.org/10.5281/zenodo.12747329 (Cai et al., 2025f).

## 6 Conclusions


In this study, we developed a comprehensive approach to estimate the spatiotemporal distribution of forest AGB across China by integrating multisource datasets—including spectral, climatic, topographic, and tree cover data—within a deep learning framework. Using over 50,000 GEDI sample points, we trained a ResNet model to produce the China Forest AGB Time Series Dataset (CFATD), a 30-meter resolution dataset spanning 1985 to 2023, accompanied by uncertainty estimates.
Validation with the 2019–2021 GEDI dataset ($R^2$ = 0.92, RMSE = 16.06 Mg/ha, Bias = 0.06 Mg/ha) and multi-year ground survey data ($R^2$ = 0.63, RMSE = 68.26 Mg/ha, Bias = -19.87 Mg/ha) confirmed the robustness of the AGB time series maps, which effectively capture the dynamics of forest biomass. The CFATD not only aligns with existing products in spatial consistency but also provides an improved representation of fine-scale spatial heterogeneity and temporal trends. Our nationwide AGB estimates revealed that as of 2023, China's average forest AGBD was 122.69±13.94 Mg/ha. The regions
with the highest AGB carbon densities were concentrated in the Southwest, Northwest, Northeast, and Taiwan. The total forest AGC stock was estimated at 13.97±0.87 PgC, with 41.5% concentrated in five provinces, highlighting the uneven distribution of forest resources. Between 1985 and 2023, average AGBD increased from 91.58±9.27 Mg/ha to 122.69±13.94 Mg/ha, and total carbon stock rose from 5.50±0.23 PgC to 13.97±0.87 PgC, with an annual net accumulation rate of 0.22 PgC. We found that 65.1% of increase AGC stock was due to forest growth, with the remaining 34.9% attributed to forest expansion. In terms
of carbon emissions, tree cover loss contributed 47.8% of AGC reductions, while forest loss (forest-to-non-forest transitions) accounted for 52.2%. The CFATD fills a critical gap by providing the first long-term, high-resolution forest AGB dataset for China. This dataset will be crucial for monitoring forest carbon dynamics, assessing climate change impacts, and guiding forest management and conservation policies.

## Author contributions

YC and X.Liu designed the research. YC conducted the research and wrote the draft of the paper. X.Liu, P.Zhu and X.Li supervised the research. All co-authors reviewed and revised the paper.



**Competing interests**

At least one of the (co-)authors is a member of the editorial board of Earth System Science Data.

**Acknowledgments**

We would like to thank the Earth Engine Uplift Support Team for granting the storage quota, which has greatly facilitated the progress of this research.

**Financial support**

This work was supported in part by the National Science Foundation for Distinguished Young Scholars of China under Grant 42225107; in part by the National Key Research and Development Program under Grant 2022YFB3903402; in part by
the Key Program of the National Natural Science Foundation of China 42330507; and in part by the National Natural Science Foundation of China under Grant 42171409, and Grant 42171410.

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
