# Peer review of "Dynamics of China's Forest Carbon Storage: The First 30 m Annual Aboveground Biomass Mapping from 1985 to 2023"

_Earth System Science Data, 2025_

## Referee Comment (RC1)

This study combines multi-source remote sensing data with residual neural networks (ResNets) to develop the first 30 m resolution annual China Forest AGB dataset (1985–2023) with uncertainty quantification. It's a long-term series of AGB data, which is of great significance for estimating forest carbon sink in China.

Comments:

**1. Introduction**

- The first three paragraghs need to be condensed.
- Line 90-100, It is best to compare the advantages and disadvantages of the existing data. eg: Have you compared international time series data, such as the AGB data of the European Space Agency, etc.
- Why did you choose the ResNet-based deep learning algorithm? You should make it clear, because this is your core technical approach.

**2. Methodology & Data**

- The GEDI AGBD samples are between 2019 to 2021, and field survey data across China is from the 1990s to the 2010s. How did you deal with the gap in time.
- Given that GEDI data has only been available since 2019, how to ensure the model's predictive ability for the early years (1985-2018) requires a more detailed explanation.

**3. Results**

- **Spatial pattern of forest AGB:** Figure 6 shows the spatial pattern of AGB in 2023, but it lacks corresponding diagrams for earlier years (such as 1985), making it difficult to visually compare the changes. It is suggested to supplement the AGB distribution map of key years.
- **Trends in AGBD/AGC**: Critique the reported increase in AGBD (95.74 to 122.69 Mg/ha) and AGC stock (5.50 to 13.97 PgC). Are these trends plausible given China's afforestation efforts?
- **Drivers of Change**: Evaluate the partitioning of AGC growth (65.1% forest growth vs. 34.9% expansion). Are these proportions supported by ancillary data (e.g., land-use maps)?

**4. Discussion**

- You compared trends of different datasets, your data has a long time series, is there more accurate for every year? I think you'd better use field data to verify.
- Although the comparison with GEDI data showed a smaller deviation (0.06 Mg/ha), the comparison with ground survey data showed a larger negative deviation (-19.87 Mg/ha), especially in the tropical rainforest area (-68.57 Mg/ha). This systematic deviation requires more in-depth analysis and discussion.
- Although the paper discusses the influence of different canopy coverage thresholds on AGB estimation in lines 547-555, it does not quantitatively explain the degree of influence of using different thresholds (such as 10% vs. 20%) on the main conclusions of this study. It is suggested to add sensitivity analysis.

---

## Author Comment (AC2)

Responses to Reviewers' Comments for Manuscript essd-2025-96

**Dynamics of China's Forest Carbon Storage: The First 30 m Annual Aboveground Biomass Mapping from 1985 to 2023**

Addressed Comments for Publication to

*Earth System Science Data*

by

Yaotong Cai, Peng Zhu, Xing Li, Xiaoping Liu, Yuhe Chen, Qianhui Shen, Xiaocong Xu, Honghui Zhang, Sheng Nie, Cheng Wang, Jia Wang, Bingjie Li, Changjiang Wu, and Haoming Zhuang

**Response Letter**

Dear Editors,

Please find enclosed the revised version of our previous submission entitled *"Dynamics of China's Forest Carbon Storage: The First 30 m Annual Aboveground Biomass Mapping from 1985 to 2023"* with manuscript number essd-2025-96. We would like to thank you and the reviewers for the valuable comments which help improving the quality of our manuscript. In this revision, we have carefully addressed the reviewers' comments. A summary of main modifications and a detailed point-by-point response to the comments from Reviewers 1 to 2 (following the reviewers' order in the decision letter) are given below.

Sincerely,

Yaotong Cai, Peng Zhu, Xing Li, Xiaoping Liu, Yuhe Chen, Qianhui Shen, Xiaocong Xu, Honghui Zhang, Sheng Nie, Cheng Wang, Jia Wang, Bingjie Li, Changjiang Wu, and Haoming Zhuang

**Note:** To enhance the legibility of this response letter, all the editor's and reviewers' comments are typeset in boxes. Rephrased or added sentences are typeset in color. The respective parts in the manuscript are highlighted to indicate changes.

**Authors' Response to the Editor**

> **General Comments.** There are several products that provide high-resolution biomass and/or carbon stock information. One unique aspect of this dataset is its long-term coverage of forest carbon storage. However, based on a preliminary review, the temporal changes in carbon stock—particularly at fine spatial scales—appear to be relatively unreliable. This suggests that while the product may effectively capture spatial distribution, it is less capable of representing temporal dynamics. Please pay close attention to this issue.

**Response:** Thank you for this important comment. We fully agree that pixel-level year-to-year changes carry higher uncertainty, and we have added a four-tier validation framework to strengthen the assessment of temporal trends and communicate this limitation more clearly in the revised manuscript.

1. We aggregated our maps to province level and compared them against independent National Forest Inventory (NFI) statistics across multiple inventory cycles by converting timber volume to AGB using published BEF and wood density ranges (see Section 3.2.2 and Fig. 4). We assessed sign agreement, Theil–Sen slope differences and relative annual growth rates (see Section 3.4).

2. We compared decadal AGBD changes with independent remote-sensing proxies, including Gross primary production (GPP) and Vegetation Optical Depth (VOD), as well as global biomass maps (see Section 3.3, and Section 4.2.3). We used direction-of-change and regionalized magnitude tests rather than absolute calibration. [NEWLY ADDED]

3. Event-based validation (Before–After–Control–Impact, BACI): We conducted BACI tests for well-documented disturbances and policies (e.g., large-scale afforestation programs, and the typical forest degradation example). Our AGBD shows gradual declines in impacted areas, and enhanced post-policy growth in targeted regions relative to controls (see Section 3.3). [MODIFIED]

4. Stability/negative tests: Over pseudo-invariant old-growth reserves without recorded disturbances, interannual variability is low and trends are not significant, suggesting

minimal spurious drift. Finally, we quantified trend significance using Mann–Kendall with Theil–Sen slopes, and propagated model uncertainties through quantile regression to derive per-pixel and provincial confidence intervals (see Section 3.4). [MODIFIED]

5. We updated the Discussion to explicitly caution readers about pixel-scale interpretation and to declare limitations (see Section 4.2). [MODIFIED]

Across provinces, our product reproduces the direction and relative magnitude of long-term changes reported by NFI; event-year anomalies are captured at expected timings; and stable areas remain trend-free within uncertainty bounds. These results support the temporal credibility of our dataset, while we clearly state limitations on annual absolute accuracy at fine spatial scales.

We believe these additions can provide both (a) a comprehensive temporal trend assessment and (b) a clear statement of the limitations.

*Where modified in manuscript:*

*Methods: Page 4, Lines 130–140; Page 7, Lines 195–205. Results: Added a Section for evaluation of estimated AGBD trend (Pages 15–16, Lines 360-375). Discussion: Pages 29-30, Lines 585–625.*

**Authors' Response to Reviewer 1**

> **General Comments.** This study combines multi-source remote sensing data with residual neural networks (ResNets) to develop the first 30 m resolution annual China Forest AGB dataset (1985–2023) with uncertainty quantification. It's a long-term series of AGB data, which is of great significance for estimating forest carbon sink in China.

**Response:** We sincerely thank the reviewer for the constructive comments and recognition of this work. All comments have been thoroughly considered and incorporated into the revised manuscript through targeted modifications. The corresponding revisions have been clearly marked for ease of reference.

**Comment 1**

The first three paragraghs need to be condensed.

**Response:** We thank the reviewer for this helpful suggestion. We have condensed the first three paragraphs of the Introduction to improve focus and readability, while retaining the key context. The redundant background material has been shortened and moved to the Supplementary Materials where appropriate.

> *Introduction — Page 1, Lines 33–65 (revised):*
>
> *Aboveground biomass (AGB) is a critical metric for quantifying carbon stocks in forest ecosystems, representing approximately 70%–90% of total forest biomass (Cairns et al., 1997). Variations in AGB over time directly shape the dynamics of the global carbon cycle. Empirical studies estimate that forest AGB accounts for approximately 15% to 30% of the terrestrial carbon pool (Davidson and Janssens, 2006) and has sequestered an estimated 106.9 PgC over the past three decades, corresponding to roughly 17% of global fossil fuel emissions (Pan et al., 2024). Given its critical role in the global carbon cycle, AGB has been incorporated into the compliance framework of the Paris Agreement and recognized as a key monitoring indicator under the United Nations Sustainable Development Goals (SDGs) (Baccini et al., 2012; Xu et al., 2021).*
>
> *China ranks fifth globally in total forest area and possesses the largest expanse of planted forests worldwide (FAO, 2020) (see Text S1 for details on the study area). Since the 1970s, the Chinese government has launched a series of large-scale forestry initiatives aimed at improving ecological quality and mitigating environmental degradation (Liu et al., 2008; Lu et al., 2018). These programs have driven extensive afforestation and reforestation, expanding forest cover from*

*12.98% in the 1980s to 24.02% in 2021 (Fig. S1) (National Forestry and Grassland Administration, 2019; Cai et al., 2024). Between 1990 and 2019, the carbon sink capacity of global temperate forests increased by 30.2%, with China's reforestation programs contributing substantially to this rise (Pan et al., 2024). Nevertheless, the degree to which these afforestation initiatives have augmented AGB, especially regarding the spatiotemporal dynamics of carbon accumulation across China's forested landscapes, remains inadequately quantified.*

**Comment 2**

Line 90-100, It is best to compare the advantages and disadvantages of the existing data. eg: Have you compared international time series data, such as the AGB data of the European Space Agency, etc.

**Response:** Thank you for this suggestion. We have expanded the introduction to summarize the main strengths and weaknesses of representative products (ESA CCI, and GEOCAR-BON/GlobBiomass, GEDI, etc.), focusing on their spatial/temporal coverage, typical input data (optical, radar, LiDAR-based upscaling), known biases (e.g., optical saturation in high-biomass regions, coarse-resolution smoothing), and suitability for temporal analysis. A short summary sentence has been added to the Introduction (Page 4, Line 100–115).

*Introduction — Page 4, Lines 100–115 (expanded).*

*Representative global and China time series AGB products (see Supplement Text S2) exhibit complementary strengths but also systematic weaknesses that limit their direct applicability for high-fidelity, temporally explicit mapping in China. For example, coarse global products and compilations (e.g., European Space Agency's Climate Change Initiative (ESA CCI) biomass dataset, and GEOCARBON/GlobBiomass) provide extensive spatial coverage and useful baselines but tend to smooth local detail because of coarse native resolutions and multi-year compositing; some pan-tropical maps also exhibit systematic positive or negative biases outside the wet tropics (Liu et al., 2015; Santoro et al., 2021). Meanwhile, emerging spaceborne lidar (e.g., GEDI/other lidar missions) provides higher-quality vertical structure information that can reduce saturation and structural ambiguity, but footprint sparsity requires careful upscaling to wall-to-wall maps (Duncanson et al., 2022). In short, available global products trade off spatial detail, temporal continuity, and structural sensitivity — none simultaneously deliver the high spatial resolution, dense temporal coverage, and unbiased response across the full biomass gradient that are needed for national-scale, temporally consistent AGB time series in China.*

> **Comment 3**
>
> Why did you choose the ResNet-based deep learning algorithm? You should make it clear, because this is your core technical approach.

**Response:** We thank the reviewer for raising this important point. We have added a focused justification for the choice of ResNet in the Introduction and Section 2.3. Briefly: (1) its residual connections facilitate stable training and alleviate vanishing gradients when learning complex, non-linear relationships between multitemporal spectral features and biomass; (2) replacing 2D layers with 1D counterparts better matches the input structure (per-pixel multitemporal feature vectors) and reduces parameter count; (3) in our baseline comparisons, ResNet demonstrated more stable cross-validation performance and superior generalization than Random Forest and XGBoost (summary of baseline comparisons added to Supplementary Text Table 3); and (4) by effectively capturing subtle spectral–structural variations in high-biomass regions, ResNet offers the potential to mitigate spectral saturation effects inherent to single-source optical data.

*Introduction — Page 3, Lines 85–95 (brief added sentence).*

*...... However, a key challenge in predicting AGB from Landsat spectra is spectral saturation, wherein reflectance values become insensitive to biomass variations beyond certain thresholds, particularly in dense forest conditions (Zhang et al., 2019). This phenomenon restricts the capacity of traditional regression-based models to accurately capture biomass dynamics in high-biomass regions. Previous studies indicate that deep learning approaches outperform traditional machine learning algorithms in terms of cross-validation stability and generalization (Dixon et al., 2025; He et al., 2016; Huy et al., 2022). In particular, the ResNet architecture—through its residual connections—facilitates the training of deeper networks and the extraction of complex, non-linear relationships between spectral–structural features and AGB (He et al., 2016). This capability offers a promising pathway to mitigating the saturation effects inherent to single-source data.*

*Methods, Section 2.3.1 — Page 7, Lines 205–225 (expanded).*

*Figure 1 illustrates the architecture of the ResNet employed for AGBD estimation. Building on the original ResNet framework (He et al., 2016), all 2D convolutional layers were replaced with 1D counterparts to match the per-pixel multitemporal spectral feature vectors used as input and to substantially reduce the number of trainable parameters. The network comprises two residual blocks, each containing a 1D convolutional layer, batch normalization, and a Rectified Linear Unit (ReLU) activation. Each block incorporates a skip connection that adds the block's input directly to its output feature map. This residual connection stabilizes training, alleviates the vanishing gradient problem, and enables the model to learn complex, non-linear relationships between multitemporal spectral–structural features and biomass. By fusing shallow and deep feature representations, the residual mechanism also enhances the model's ability to capture subtle spectral variations in high-biomass regions, thereby helping to mitigate the saturation effects inherent to single-source*

*optical data. Following each residual block, max-pooling layers reduce the dimensionality of intermediate feature maps by half, decreasing memory consumption, accelerating inference, and gradually enlarging the receptive field so that deeper layers encode broader contextual information. To further improve generalization, a dropout layer with a rate of 0.5 is applied before the final fully connected component, which consists of two linear layers separated by a ReLU activation.*

**Comment 4**

The GEDI AGBD samples are between 2019 to 2021, and field survey data across China is from the 1990s to the 2010s. How did you deal with the gap in time.

**Response:** We appreciate this concern. We clarify that GEDI samples (2019–2021) were used exclusively for model training, while historical ground surveys (1990s–2010s) and provincial NFIs were used only for independent validation. Therefore, no direct temporal alignment was required between GEDI footprints and field surveys. We suspect that the reviewer's question closely relates to the broader issue of temporal transferability (i.e., applying a model trained with recent GEDI data to earlier Landsat imagery), which we address in detail in response to Comment 5. Briefly, we addressed this as follows: (1) spectral harmonization across Landsat sensors was applied to ensure long-term consistency; (2) each GEDI footprint was aligned to Landsat observations by matching the GEDI acquisition date to the corresponding Landsat annual composite (i.e., the Landsat mosaic nearest in time), so the ResNet learns the spectral–AGB relationship at specific acquisition times; (3) we assume that the spectral–AGB relationship generalizes backward in time when applied to earlier Landsat imagery; (4) to test this, we performed independent validation against historical ground plots (1978–2008) and provincial NFI-derived AGC.

**Comment 5**

Given that GEDI data has only been available since 2019, how to ensure the model's predictive ability for the early years (1985-2018) requires a more detailed explanation.

**Response:** Thank you for this suggestion. We expanded the Methods and Results to explain why the ResNet model trained on 2019–2021 GEDI samples can be applied to earlier years and how we validated this application. First, this approach follows a space-for-time substitution rationale: GEDI footprints sample a wide range of forest types, ages, and biomass conditions across China, allowing the model to learn spectral–biomass relationships that generalize temporally. Second, all Landsat inputs were cross-sensor harmonized to ensure spectral consistency over time, reducing spurious temporal trends due to sensor differences ([Revision] see Methods 2.1.1, Page 5, Lines 140–142). Third, the model leverages not only contemporaneous spectral inputs but also ancillary predictors (climate variables, DEM-derived topography, latitude/longitude, and annual tree cover from CATCD), which stabilize predictions over time. Finally, we evaluated temporal transfer explicitly through independent validation against historical ground plots (1978–2008) and provincial NFI trends. These results ($R^2$, RMSE, and bias by ecoregion) confirm that the model reproduces plausible historical dynamics (see Results and Page 14, Lines 330–335). As a precaution, we also expanded the Discussion to note limitations and best-use recommendations (see Page 29, Lines 585–625).

**Comment 6**

Spatial pattern of forest AGB: Figure 6 shows the spatial pattern of AGB in 2023, but it lacks corresponding diagrams for earlier years (such as 1985), making it difficult to visually compare the changes. It is suggested to supplement the AGB distribution map of key years.

**Response:** Thank you for this helpful suggestion. We have updated Figure 6 (see Page 20, Figure 9) to include additional panels for earlier benchmark years (1985, 2000 and 2015) to

allow direct visual comparison of spatial AGB patterns across time. Due to space constraints the full annual series is provided in the Supplement (Supplementary Fig. S5). The Figure 6 (Figure 9 in revised manuscript) caption and corresponding paragraph in Section 3.3 (Section 3.4 in revised manuscript) have been updated to reference these additions.

**Comment 7**

Trends in AGBD/AGC: Critique the reported increase in AGBD (95.74 to 122.69 Mg/ha) and AGC stock (5.50 to 13.97 PgC). Are these trends plausible given China's afforestation efforts?

**Response:** We appreciate the reviewer's request for critical evaluation. The observed increase in mean AGBD and AGC is consistent with China's large-scale afforestation and forest rehabilitation programs since the 1980s and especially since the late 1990s (e.g., Grain-for-Green). Our analysis partitions the total AGC increase into forest growth (internal stand accretion) and forest expansion (area increase), and independent comparisons with provincial NFI totals and several external products show broadly consistent upward trends (see Fig. 4 and Fig. 16). Our findings are also consistent with previous studies, confirming that China's forests have served as a significant carbon sink over the past few decades (see Supplementary Text S4).

At the same time, we acknowledge potential sources of systematic uncertainty that may affect the magnitude: (1) optical indices tend to saturate in very high-biomass forests which may bias estimates low in tropical/rainforest regions; (2) forest definition (we used 20% tree-cover threshold) influences total forested area and thus AGC; (3) the biomass-to-carbon conversion factor (0.5) is a simplification that ignores species- and region-specific wood carbon fractions. We have expanded the Discussion to explicitly discuss these caveats and their potential quantitative effects (see Discussion, Page 29, Lines 585–625).

**Comment 8**

Drivers of Change: Evaluate the partitioning of AGC growth (65.1% forest growth vs. 34.9% expansion). Are these proportions supported by ancillary data (e.g., land-use maps)?

**Response:** Yes, our partitioning is supported by the China Annual Tree Cover Dataset (CATCD), which indicates that between 1985 and 2023, 67% of forest increase was due to canopy closure (growth) and 33% to expansion, closely matching our AGC partitioning. By contrast, the LULC dataset is binary and cannot separate these two processes.

**Comment 9**

You compared trends of different datasets, your data has a long time series, is there more accurate for every year? I think you'd better use field data to verify.

**Response:** Thank you for this valuable suggestion. To address the your concern and despite the absence of China-wide permanent biomass plots with annual revisits, we have added a four-tier validation framework to strengthen the assessment of temporal trends in the revised manuscript.

1. We aggregated our maps to province level and compared them against independent National Forest Inventory (NFI) statistics across multiple inventory cycles by converting timber volume to AGB using published BEF and wood density ranges (see Section 3.2.2 and Fig. 4). We assessed sign agreement, Theil–Sen slope differences and relative annual growth rates (see Section 3.4).

2. We compared decadal AGBD changes with independent remote-sensing proxies, including Gross primary production (GPP) and Vegetation Optical Depth (VOD), as well as global biomass maps (see Section 3.3, and Section 4.2.3). We used direction-of-change and regionalized magnitude tests rather than absolute calibration. [NEWLY ADDED]

3. Event-based validation (Before–After–Control–Impact, BACI): We conducted BACI tests for well-documented disturbances and policies (e.g., the 2008 southern China ice storm,

major typhoon landfalls, and large-scale afforestation programs). Our AGBD shows immediate declines and gradual recovery in impacted areas, and enhanced post-policy growth in targeted regions relative to controls (see Section 3.3.2). [MODIFIED]

4. Stability/negative tests: Over pseudo-invariant old-growth reserves without recorded disturbances, interannual variability is low and trends are not significant, suggesting minimal spurious drift. Finally, we quantified trend significance using Mann–Kendall with Theil–Sen slopes, and propagated model uncertainties through quantile regression to derive per-pixel and provincial confidence intervals (see Section 3.4).

Across provinces, our product reproduces the direction and relative magnitude of long-term changes reported by NFI; event-year anomalies are captured at expected timings; and stable areas remain trend-free within uncertainty bounds. These results support the temporal credibility of our dataset, while we clearly state limitations on annual absolute accuracy at fine spatial scales.

*Where modified in manuscript:*
*Methods: Page 4, Lines 130–140; Page 7, Lines 195–205. Results: Added a Section for evaluation of estimated AGBD trend (Pages 15–16, Lines 360-375). Discussion: Pages 29-30, Lines 585–625.*

**Comment 10**

Although the comparison with GEDI data showed a smaller deviation (0.06 Mg/ha), the comparison with ground survey data showed a larger negative deviation (-19.87 Mg/ha), especially in the tropical rainforest area (-68.57 Mg/ha). This systematic deviation requires more in-depth analysis and discussion.

**Response:** We thank the reviewer for highlighting this important point. We conducted an in-depth residual analysis and found several factors contributing to the larger negative bias against historical ground surveys, particularly in tropical rainforests: (1) scale mismatch, as field plots often exceed the 30 m pixel footprint and thus differ in representativeness; (2) spectral saturation in dense canopies, which reduces sensitivity of optical predictors at very high

AGBD and high tree cover; (3) potential transferability limitations of the GEDI AGBD product in certain tropical regions; and (4) differences in allometric models or biomass estimation protocols used in historical surveys. To illustrate these issues, we examined residual patterns against observed AGB, tree cover, and slope (Fig. 5), and we expanded the Discussion (Page 28, Lines 585–625) to further analyze the causes and outline recommendations for future improvement.

*Results: detailed residual diagnostics — Page 14, Lines 340–355.*

*The comparison between our AGBD estimates and historical ground surveys (Fig. 3) indicates a systematic underestimation in some regions, as reflected by regression slopes lower than one and a negative mean bias. Additional diagnostics (Supplementary Table S3) show that this bias increases with observed biomass density, implying greater underestimation in high-biomass plots. This pattern is consistent with the well-known spectral saturation effect, whereby optical indices lose sensitivity in dense canopies, as well as with potential scale and methodological mismatches between satellite-based estimates and historical field plots.*

*The residual analysis stratified by geographic location, tree cover, and terrain further underscores these tendencies (Fig. 5). Residuals are wider in low latitude (<25°N) and high latitude (>40°N) regions compared to mid-latitudes. As slope and tree cover increase, residual spread also broadens, indicating that predictions tend to be less accurate in steep or densely forested areas. By contrast, no significant biases were observed across longitudes, aspects, or elevations, suggesting that errors are more strongly associated with biomass density and slope rather than with all terrain factors. These findings highlight slope- and biomass-related challenges that should be addressed in future AGBD mapping efforts.*

*Discussion: extended analysis and recommendations — Page 28, Lines 585–625.*

*Current AGBD maps often overestimate low values and underestimate high values—a persistent pattern well documented in previous studies (Réjou-Méchain et al., 2019; Rodríguez-Veiga et al., 2019). These discrepancies are largely driven by sensor limitations, such as the saturation of optical and radar signals in high-biomass forests, leading to systematic underestimation in dense tropical or subtropical stands. The overestimation of low AGBD is more complex, often arising from regression models fitted to limited or non-representative plot data. Additional systematic biases can result from measurement errors, incorrect allometric models, and environmental factors (e.g., precipitation, snow cover) that influence remote sensing signals (Santoro et al., 2021). Furthermore, mismatches between the spatial resolution of remote sensing data and the size or geolocation accuracy of field plots can distort pixel-level estimates, underscoring the need for improved plot–pixel matching and more representative field measurements.*

*Against this backdrop, our estimates indicate a substantial increase in mean forest AGBD from 95.74 to 122.69 Mg ha-1 and in total AGC stock from 5.50 to 13.97 PgC over the study period. These upward trends are broadly consistent with the scale and trajectory of China's large-scale afforestation and forest rehabilitation programmes since the 1980s—especially post-1999 initiatives such as the Grain-for-Green Program—which have simultaneously expanded forest area and increased stand biomass through natural growth and management. Independent comparisons with provincial NFI statistics and several external products show similar upward trajectories (see Figs. 4 and 15), lending confidence to the observed direction of change. Nevertheless, we acknowledge several sources of systematic uncertainty that could affect the magnitude of these increases. First, optical indices tend to saturate in very high-biomass forests, potentially biasing*

*estimates downward in tropical and old-growth stands. Second, our definition of forest ($\geq$ 20 % tree-cover threshold) directly influences the mapped forest extent and thus the total AGC stock; a higher threshold would yield lower total area and carbon estimates (see supplementary Text S5). Third, the use of a fixed biomass-to-carbon conversion factor (0.5) ignores species- and region-specific variation in carbon content, which may introduce bias in regional and national totals. Finally, the GEDI footprint AGBD dataset, which informs part of our modelling framework, lacks direct training data from China and neighbouring countries such as Russia (Duncanson et al., 2022). Although the Level 4A algorithm is designed for geographic transferability (Kellner et al., 2023), the absence of local calibration data may still introduce uncertainty in parts of China.*

*While deep learning approaches have been employed here to reduce spectral saturation effects and improve estimation accuracy, further advances are possible. Integrating multi-source remote sensing data (optical, radar, LiDAR) can better capture forest structural complexity and mitigate the limitations of individual sensors (see supplementary Text S6). Developing region- and forest-type-specific allometric models will further refine biomass estimation in heterogeneous landscapes. Expanding calibration and validation with more extensive and diverse ground data, particularly from underrepresented regions, will improve both accuracy and transferability. Incorporating time-series data into modelling frameworks can also help distinguish between short-term fluctuations and long-term growth trends (Araza et al., 2022; Duncanson et al., 2019).*

*To guide proper use of these maps, we note several best-use recommendations: the products are most reliable at regional to national scales rather than for individual plots; caution is warranted when interpreting high-biomass tropical or subtropical forests due to potential saturation; and users should consider forest definition thresholds and local conversion factors when estimating carbon stocks. Where possible, independent validation with local field data is recommended before applying these maps for management or policy decisions. Finally, continued development of deep learning models and integration of multi-source data are likely to further enhance both the accuracy and applicability of future AGBD products.*

**Comment 11**

Although the paper discusses the influence of different canopy coverage thresholds on AGB estimation in lines 547-555, it does not quantitatively explain the degree of influence of using different thresholds (such as 10% vs. 20%) on the main conclusions of this study. It is suggested to add sensitivity analysis.

**Response:** We appreciate this suggestion. We performed a sensitivity analysis using tree-cover thresholds of 10%, 20% (main text), and 30%. The results are summarized in Supplementary Text S5 and briefly discussed in the Discussion section (Page 28, Line 585-625). As expected, absolute AGC totals vary with the chosen threshold, but the relative temporal trends (overall

increase in mean AGBD and a net national AGC sink) remain consistent across thresholds. At lower thresholds (10% and 20%), forest growth accounts for the majority of AGC gains (80.81% and 65.06%, respectively), whereas at a higher threshold (30%), forest expansion slightly exceeds growth (52.79% vs. 47.21%). Quantitative differences in national AGC and the contributions of each change mode are reported in Supplementary Text Table 5 for transparency.

*Where modified in manuscript:*

*Supplementary materials: Text S5 [NEWLY ADDED]; Discussion: Pages 29-30, Lines 585–625 [MODIFIED].*

**Authors' Response to Reviewer 2**

> **General Comments.** This manuscript generated a long-term 30 m Annual Aboveground Biomass data from 1985 to 2023 in China. They mainly used a residual neural networks (ResNets) together with the GEDI footprint AGBD and Landsat images to generate the AGBD map. The manuscript is generally well-written. I have several suggestions listed below, which I think may be helpful for revising the manuscript.

**Response:** We sincerely thank the reviewer for the positive evaluation of our work and for recognizing the contribution of our study. We also greatly appreciate the constructive suggestions provided, which help us further improve the clarity, rigor, and completeness of the manuscript. Detailed, point-by-point responses to each suggestion are provided below, along with the corresponding revisions made in the manuscript.

> ### Comment 1
> Does the observations from Landsat suffers from the sensor degradation effect? Since they are merged by different sensors, it may have some impacts on the long-term trend of Landsat data, which can finally propagate to the AGBD map.

**Response:** We fully agree with the reviewer's observation. To address this, we used Landsat Collection 2 Level-2 Tier-1 surface reflectance products, which include sensor-specific radiometric calibration and atmospheric correction. These data are continuously recalibrated by the USGS to compensate for performance degradation due to mechanical, electronic, or ultraviolet effects, thereby maintaining the geometric, radiometric, spatial, and spectral standards of the archive. Annual mediod composites were generated with rigorous QA filtering to minimize sensor noise, cloud contamination, and other artefacts. To harmonize reflectance from different Landsat sensors, we followed the spectral harmonization procedure described by Roy et al. (2016), reducing potential long-term trend biases attributable to sensor transitions. Collectively,

these pre-processingg steps minimize the likelihood of consistent long-term bias in our AGBD time series, as detailed in the revised Methods section (Page 5, Line 130–140).

*Methods — Page 5, Lines 130–140 (Landsat processing and harmonization described).*

*This study utilized Landsat satellite imagery from the Google Earth Engine (GEE) platform, covering the period from 1985 to 2023. The dataset includes images from Landsat 4/5 Thematic Mapper (TM), Landsat 8/9 Operational Land Imager (OLI), and Landsat 7 Enhanced Thematic Mapper Plus (ETM+) prior to 2003. Due to the Scan Line Corrector (SLC)-off failure in mid-2003, post-2003 Landsat 7 data were excluded from the analysis, except for 2012 when no other Landsat sensors provided coverage. All imagery was obtained from the Landsat Collection 2, Level-2, Tier-1 surface reflectance products at 30 m spatial resolution, which have undergone sensor-specific radiometric calibration, atmospheric correction, and orthorectification (Wulder et al., 2022).*

*To ensure temporal consistency, we first selected images acquired during the main growing season (Day of Year 150–300) and applied the cross-sensor spectral harmonization procedure described by Roy et al. (2016) to align the spectral responses of TM, ETM+, and OLI bands. Subsequently, annual medoid composites were generated for each year using QA band filtering (Zhu et al., 2015) to remove cloud, shadow, and snow contamination. Remaining gaps in the annual composites were filled using linear interpolation or extrapolation along the temporal dimension. Each composite contained three visible bands (red, green, blue), one near-infrared (NIR) band, and two shortwave infrared (SWIR) bands (Table 1), which served as input features for subsequent modelling. A more detailed description of the processing workflow can be found in Cai et al. (2025).*

**Comment 2**

How do you consider the risk of overfitting of your model? And the multicollinearity issue among the predictors you choose?

**Response:** Thank you for these questions. To mitigate overfitting, we employed recursive feature elimination (RFE) with 5-fold cross-validation to select a compact predictor set (final 17 variables), applied dropout (rate = 0.5) during ResNet training, and evaluated generalization on a withheld test set comprising 20% of the GEDI samples (see Section 2.4.1, Page 7, Lines 209–213). Model performance on GEDI AGBD samples in both the training and testing sets indicated no significant signs of overfitting (see Table 1).

Regarding multicollinearity, the RFE procedure inherently reduces redundancy by removing highly correlated predictors. Moreover, deep learning models are generally robust to multicollinearity because they learn hierarchical, non-linear representations of the input space rather than relying solely on mutually independent features [1], [2].

[1]  R. D. De Veaux and L. H. Ungar, "Multicollinearity: A tale of two nonparametric regressions," in *Selecting Models from Data*, P. Cheeseman and R. W. Oldford, Eds., New York, NY: Springer New York, 1994, pp. 393–402.

[2]  A. Hamedianfar, C. Mohamedou, A. Kangas, and J. Vauhkonen, "Deep learning for forest inventory and planning: A critical review on the remote sensing approaches so far and prospects for further applications," *Forestry: An International Journal of Forest Research*, vol. 95, no. 4, pp. 451–465, Feb. 2022. DOI: 10.1093/forestry/cpac002. eprint: https://academic.oup.com/forestry/article-pdf/95/4/451/45293980/cpac002.pdf.
* * *
**Comment 3**

How do you validate the long-term trend of AGBD you generated? Since it is the key novelty aspect of this study.
* * *
**Response:** Thank you for this question. We implemented a four-tier validation framework to evaluate the credibility of interannual and multi-decadal AGBD changes.

1. We aggregated our maps to province level and compared them against independent National Forest Inventory (NFI) statistics across multiple inventory cycles by converting timber volume to AGB using published BEF and wood density ranges (see Section 3.2.2 and Fig. 4). We assessed sign agreement, Theil–Sen slope differences and relative annual growth rates (see Section 3.4).

2. We compared decadal AGBD changes with independent remote-sensing proxies, including Gross primary production (GPP) and Vegetation Optical Depth (VOD), as well as global biomass maps (see Section 3.3, and Section 4.2.3). We used direction-of-change and regionalized magnitude tests rather than absolute calibration. [NEWLY ADDED]

3. Event-based validation (Before–After–Control–Impact, BACI): We conducted BACI tests for well-documented disturbances and policies (e.g., the 2008 southern China ice storm, major typhoon landfalls, and large-scale afforestation programs). Our AGBD shows immediate declines and gradual recovery in impacted areas, and enhanced post-policy growth in targeted regions relative to controls (see Section 3.3.2). [MODIFIED]

4. Stability/negative tests: Over pseudo-invariant old-growth reserves without recorded disturbances, interannual variability is low and trends are not significant, suggesting minimal spurious drift. Finally, we quantified trend significance using Mann–Kendall with Theil–Sen slopes, and propagated model uncertainties through quantile regression to derive per-pixel and provincial confidence intervals (see Section 3.4).

Across provinces, our product reproduces the direction and relative magnitude of long-term changes reported by NFI; event-year anomalies are captured at expected timings; and stable areas remain trend-free within uncertainty bounds. These results support the temporal credibility of our dataset, while we clearly state limitations on annual absolute accuracy at fine spatial scales.

*Where modified in manuscript:*

*Methods: Page 4, Lines 130–140; Page 7, Lines 195–205. Results: Added a Section for evaluation of estimated AGBD trend (Pages 15–16, Lines 360-375). Discussion: Pages 29-30, Lines 585–625.*
* * *
**Comment 4**

Figure 3: from this figure, it suggests that your AGBD map is systematically lower than the observed value, right? Since the slope is lower than one, and with a negative bias value. Please explain this point.
* * *
**Response:** Yes, as the reviewer correctly noted, Figure 3 shows a negative bias and slopes < 1 in some ecoregions. We conducted additional diagnostics (Supplementary Text Table 6), which confirm that the bias increases with observed biomass, i.e., underestimation is greater in high-biomass plots. This pattern is attributable to spectral saturation in dense canopies and

potential scale/method mismatches between satellite estimates and historical plot inventories. We have expanded the Results (Section 3.2.3, Page 14, Lines 340–350) to explicitly interpret these patterns and added a discussion of potential corrections (Page 30, Lines 610–615), such as incorporating SAR or airborne LiDAR to improve accuracy in high-biomass regions.

*Where modified in manuscript:*
*Results: Section 3.2.3, Page 14, Lines 340–350 (Expanded). Discussion: Page 30, Lines 610–615 (Expanded).*

**Comment 5**

Figure 8: how do you validate that the temporal trend in AGBD you generated is correct or not? Could you find some ground observations to validate this?

**Response:** We appreciate this valuable comment. We kindly refer to our detailed response to Comment 3, where we have explained the validation strategy and supporting evidence.

**Comment 6**

Figure 9: how do you separate the AGBD trend into forest expansion and forest growth? Please added some detailed process of how you did this.

**Response:** We thank the reviewer for this important question. We would like to clarify that our attribution analysis was conducted on AGC changes between 1985 and 2023, not directly on AGBD. We suspect that the reviewer may have intended to refer to AGC attribution rather than AGBD. The detailed procedure is now explicitly described in Methods 2.8 (Page 10, Lines 270-290). In brief: (1) land-cover transitions were classified using CATCD with a 20% tree-cover threshold; (2) per-pixel AGC change ($\triangle$AGC) was computed between 1985 and 2023; (3) $\triangle$AGC in persistent-forest pixels was attributed to forest growth (positive) or tree cover loss (negative); (4) $\triangle$AGC in pixels transitioning from non-forest to forest was

attributed to forest expansion; and (5) $\triangle$AGC in pixels transitioning from forest to non-forest was attributed to forest loss.

*Where modified in manuscript:*
*Methods: Page 10, Lines 270–290 (EXPANDED).*

**Comment 7**

Figure 15: we can find a large difference (even two-three times larger) between your map with Hengeveld, CCI and Chen's maps. Could you explain the underlying reason?

**Response:** Thank you for pointing this out. We have expanded the Discussion around Fig. 16 to explain the underlying reasons for the discrepancies. Specifically, we now clarify that the differences mainly stem from: (1) spatial resolution (global products at $\geq$1 km smooth local dynamics, whereas our 30 m dataset captures finer variations); (2) varying forest definitions and masks, which alter the mapped forest extent; (3) differences in input datasets and modeling methods (e.g., radar/LiDAR upscaling vs. optical-statistical approaches) that respond differently to canopy structure; and (4) temporal mismatches in training and reference samples. Importantly, many existing global AGB datasets rely on coarse LULC or forest maps that cannot capture China's large-scale afforestation programs, leading to systematic underestimation of forest gains. We also added subregional examples (Fig. 15) to illustrate how each factor contributes to divergence. These clarifications have been incorporated into the revised Discussion (Page 28, Lines 560–585 and Supplementary Text S4).

*Section 4.1.2 — Page 28, Lines 560–585 (expanded discussion).*
*Our AGBD dataset reveals a consistent upward trend from 1985 to 2023, highlighting the steady recovery and effective management of China's forests (Fig. 15). This trajectory is broadly in line with national afforestation and conservation programs, and provides a robust depiction of long-term carbon stock dynamics (Zeng et al., 2023). To benchmark our results, we compared them with four independent AGB time-series datasets (see Text S2).*
*The comparison highlights substantial differences in both magnitude and temporal pattern. For instance, the Hengeveld dataset (Hengeveld et al., 2015) exhibits pronounced fluctuations, including a decline during the late 1990s and early 2000s, while the ESA CCI product shows a general decrease in the 2010s. Chen et al. (2023) also report consistently*

*lower AGBD values than our estimates. These discrepancies arise from several key factors: (1) spatial resolution—most global products are provided at ≥1 km resolution, which smooths local heterogeneity and suppresses extremes captured by our 30 m dataset; (2) forest definition and masking—different tree-cover thresholds (e.g., Chen et al., (2023) use >15% canopy cover) alter the mapped forest extent and introduce mixed pixels, leading to lower mean AGBD. Notably, when Chen's product is aggregated with the 1 km forest cover map used by Hu et al. (2016), its average AGBD increases substantially, ranging from 68.94 Mg ha$^{-1}$ in 2002 to 75.58 Mg ha$^{-1}$ in 2021, underscoring the strong dependence of biomass estimates on forest masks. (3) input data and modeling approaches—radar- or lidar-upscaled products differ in their sensitivity to canopy structure compared with optical or inventory-driven models; (4) many existing datasets rely on coarse LULC or forest masks that cannot effectively capture China's large-scale afforestation efforts, thereby underestimating biomass gains; and (5) temporal coverage and reference data—the calibration periods and reference samples vary among products, creating temporal mismatches in trend estimation (Quegan et al., 2019; Santoro et al., 2021).*

*Despite these differences, our dataset shows stronger agreement with independent evidence from national forest inventory statistics and regional studies (Fang et al., 2001; Liu et al., 2015; Piao et al., 2005; Zeng et al., 2023), all of which support a long-term upward trajectory of China's forest biomass. This comparison further underscores the value of high-resolution, long-term mapping for capturing heterogeneous forest dynamics in China.*

---

## Author Response (AR2)

**Dynamics of China's Forest Carbon Storage: The First 30 m Annual Aboveground Biomass Mapping from 1985 to 2023**

Addressed Comments for Publication to

*Earth System Science Data*

by

Yaotong Cai, Peng Zhu, Xing Li, Xiaoping Liu, Yuhe Chen, Qianhui Shen, Xiaocong Xu, Honghui Zhang, Sheng Nie, Cheng Wang, Jia Wang, Bingjie Li, Changjiang Wu, and Haoming Zhuang

**Response Letter**

Dear Editors,

Please find enclosed the revised version of our manuscript entitled *"Dynamics of China's Forest Carbon Storage: The First 30 m Annual Aboveground Biomass Mapping from 1985 to 2023"* (Manuscript ID: essd-2025-96). We sincerely thank the editor and reviewers for their continued time and constructive feedback. We appreciate the positive assessment and the helpful minor comments provided in this round, which have further improved the quality of the manuscript. All comments have been carefully addressed, and corresponding revisions have been made in the manuscript. A concise summary of the main modifications and a detailed point-by-point response to the reviewers' comments are provided below, following the order of the decision letter.

Sincerely,

Yaotong Cai, Peng Zhu, Xing Li, Xiaoping Liu, Yuhe Chen, Qianhui Shen, Xiaocong Xu, Honghui Zhang, Sheng Nie, Cheng Wang, Jia Wang, Bingjie Li, Changjiang Wu, and Haoming Zhuang

**Note:** To enhance the legibility of this response letter, all the editor's and reviewers' comments are typeset in boxes. Rephrased or added sentences are typeset in color. The respective parts in the manuscript are highlighted to indicate changes.

**Authors' Response to the Editor**

**General Comments.** There are a few minor suggestions from the reviewers. Please consider addressing them accordingly.

**Response:** We thank the editor for the guidance. All minor suggestions from the reviewers have been carefully addressed, and corresponding revisions have been made in the manuscript.

**Authors' Response to Reviewer 1**

> **Comment 1**
>
> Thanks for these revisions. Most of my concerns have been revised. I have only one minor suggestions. As the authors also concur, the underestimation of your map is greater in high-biomass plots, and your map has a large difference with existing maps (CCI and others). I suggest the authors add some caveats in the discussion to highlight these issues, since the readers may use your data with these cautions.

**Response:** We thank the reviewer for this valuable suggestion. We have added a statement in the Discussion section to acknowledge the underestimation in high-biomass regions and the differences from existing datasets. This addition highlights the potential sources of bias and advises users to interpret the data with these considerations in mind. The revision has been made and is marked in the manuscript.

*Discussion — Page 30, Lines 620–625 (added caveats):*

*To guide proper use of these maps, we note several best-use recommendations: the products are most reliable at regional to national scales rather than for individual plots; caution is warranted when interpreting high-biomass tropical or subtropical forests due to potential saturation, and users are encouraged to cross-validate with other datasets when conducting local analyses; and users should consider forest definition thresholds and local biomass-to-carbon conversion factors when estimating carbon stocks. Where possible, independent validation with local field data is recommended before applying these maps for management or policy decisions. Finally, continued development of deep learning models and integration of multi-source data are likely to further enhance both the accuracy and applicability of future AGB products.*

**Authors' Response to Reviewer 2**

> **General Comments.** This a really good paper and the authors did a good revision. However, before publishing, some minor issues also need to be addressed.

**Response:** We thank the reviewer for the positive evaluation of our work and for pointing out the remaining minor issues. All comments have been carefully addressed, and corresponding revisions have been made in the manuscript.

**Comment 1**

Geographic location is a man-made variable and does not have physical link to AGB. Is it reasonable to include it as a predictor; I think it will not affect too much using some other predictors instead.

**Response:** We thank the reviewer for this insightful comment. We agree that geographic coordinates are not direct physical drivers of AGB. However, spatial autocorrelation is a well-established property of most environmental variables, and including coordinates helps the model capture large-scale spatial patterns and residual spatial dependence not fully explained by other predictors. In our feature selection process using the Recursive Feature Elimination (RFE) method, geographic coordinates were consistently identified as important variables, and removing them reduced predictive accuracy (Figure 2). This finding is consistent with previous studies, which have shown that incorporating spatial coordinates can improve both model stability and precision (Yang and Huang., 2021). We have clarified this rationale in the revised Methods section.

> *Methods — Page 6, Lines 170–175.*
>
> *To capture large-scale spatial patterns and residual spatial autocorrelation beyond environmental predictors, we also included latitude and longitude as auxiliary spatial variables, following Tobler's First Law of Geography. This practice has proven effective in improving model generalization and reducing spatial bias in regions with sparse training data (Yang and Huang, 2021).*

**Comment 2**

L120 and 181, the number should be consistent.

**Response:** Thank you for pointing this out. We have checked and corrected the inconsistency between the two numbers to ensure they are now consistent in the revised manuscript.

*Where modified in manuscript:*
*Introduction — Page 4, Lines 119–120.*

**Comment 3**

As a data paper, the field survey data should be given and it is important; More details should also be given how the field survey data were used for validation; because the study has a weakness for temporal validation, could you make a temporal validation using survey data for given years? See Laffitte et al., Global Change Biology, 2025

**Response:** We thank the reviewer for emphasizing the importance of data transparency. The field survey data used in this study were compiled from published sources (Avitabile et al., 2016; Luo et al., 2014; Usoltsev and , 2020; Zhang et al., 2019). These datasets were collected and curated by the respective authors and are publicly available through the cited publications. To respect data ownership and licensing agreements, we did not redistribute these field data directly but provided complete citations to their sources. Although we cannot share the data ourselves, all datasets are publicly accessible through the referenced studies, where detailed access information is provided.

Following the reviewer's suggestion, we have added more details in the Methods section (Section 2.2.2) to clarify how the field survey data were used for validation. Specifically, we used 2,109 ground plots as an independent validation dataset, separate from the GEDI AGBD samples used for model training (2019–2021). Model-predicted AGBD values were compared against field-based estimates to compute validation metrics including the coefficient

of determination (R²), root-mean-square error (RMSE), and mean bias. The results are presented in Figure 3 to assess the overall accuracy and potential bias of the predicted AGBD.

We appreciate the reviewer's suggestion regarding temporal validation. The Leave-One-Year-Out (LOYO) approach used in Laffitte et al. (2025) is an effective method to assess model generalization over time. However, this method requires multi-year and temporally consistent training data. In our case, the field survey data (1978–2008) were compiled from heterogeneous studies conducted in different years, regions, and sampling protocols, which makes them unsuitable for LOYO validation.

Therefore, we implemented the LOYO validation using the GEDI AGBD dataset (2019–2021), which provides consistent multi-temporal observations. Three annual models were trained, each time leaving one year out for validation. This procedure follows the same rationale as LOYO validation but is based on the most temporally consistent data available.

> *Where modified in manuscript:*
> *Methods: Page 9, Lines 255–260.*
> *To assess temporal robustness, we implemented a Leave-One-Year-Out (LOYO) validation using the multi-year GEDI AGBD record (2019–2021). In each LOYO iteration (i.e., leaving one year out for validation), the model was trained on the remaining two years, and the process was repeated three times to evaluate temporal generalization. In addition, a 5-fold cross-validation was performed within the full dataset to further assess model stability and ensure consistent predictive performance. Reported accuracy metrics represent the mean performance across LOYO and 5-fold evaluations. For assessing the accuracy of the AGBD estimation model, we utilized validation metrics such as the coefficient of determination (R2, Eq. (2)), root mean square error (RMSE, Eq. (3)), and bias (Eq. (4)). These metrics quantified the agreement between predicted AGBD values and reference values, providing insights into prediction accuracy and systematic errors.*
>
> *Results: Page 11, Lines 305–315.*
> *The ResNet model exhibited strong predictive performance for AGBD. In 5-fold cross-validation, the model achieved R2=0.91, RMSE = 16.49 Mg ha1, and a minimal bias of 0.50 Mg ha1, demonstrating its overall accuracy. Temporal validation using the LOYO approach yielded slightly lower but still robust performance (R2=0.85, RMSE = 21.20 Mg ha1, Bias = 0.66 Mg ha1; Fig. 3), highlighting the model's ability to generalize across years. The somewhat lower LOYO performance, particularly for 2021, reflects the limited training data from 2019–2020 relative to the 2021 prediction set, which reduced the model's exposure to the high-biomass conditions in 2021. Nevertheless, the model still delivered satisfactory predictions across all years. Moreover, the ResNet model outperformed other machine learning ensemble models (e.g., Random Forest, XGBoost, and LightGBM), particularly in mitigating the effects of spectral saturation in high-biomass forests (see Text S3 for details).*

**Comment 4**

The logic of using GPP and VOD should be further clarified.

**Response:** We thank the reviewer for the comment. We clarified in the manuscript that GPP and VOD were used to evaluate the temporal trends of predicted AGB because they provide independent, complementary signals of vegetation dynamics: GPP reflects carbon uptake, while VOD reflects canopy structure and water content, both of which are strongly linearly related to AGB. Consistent trends between predicted AGB and these variables increase confidence that the observed temporal patterns are reliable. This rationale and the analysis procedure have been added to the Methods section.

*Where modified in manuscript:*

*Materials and methods: Section 2.3, Page 7, Lines 200–201 (Expanded).*

*To cross-check the temporal dynamics of AGBD, we employed two independent datasets: vegetation optical depth (VOD) and gross primary production (GPP). VOD primarily reflects canopy structural and water characteristics, whereas GPP quantifies ecosystem carbon uptake; both are strongly and positively correlated with aboveground biomass dynamics. VOD was derived from the VOD Climate Archive (VODCA v2, CXKu band), which provides harmonized microwave retrievals from 1987 to 2021 at 0.25° resolution (Zotta et al., 2024). To better capture biomass-related signals, we filtered daily VOD by the main growing season (day of year 150–300) and composited annual medians. For ecosystem productivity, we used the Global Sunlit and Shaded GPP dataset (1992–2020, 0.05° resolution), which estimates photosynthesis with a two-leaf light use efficiency model. Annual GPP was adopted to evaluate AGBD variations at interannual to decadal scales (Bi et al., 2022). Together, VOD and GPP provide complementary perspectives on canopy structure, water content, and carbon uptake, supporting the validation of long-term AGBD trends.*

**Comment 5**

L228: do you make a bootstrap for how many times? If you just make one random selection, the model may be unrealiable.

**Response:** We thank the reviewer for this insightful comment. To ensure model robustness, we included a 5-fold cross-validation scheme, in which 80% of the samples were used for training and 20% for validation in each iteration. The process was repeated five times with different random partitions, and the mean and standard deviation of model performance metrics across

folds were used to assess model stability. This revision has been added to the Methods and Results section.

*Where modified in manuscript:*

*Materials and methods: Section 2.5, Page 6, Lines 259–260; Results: Section 3.2.1, Page 11, Lines 305–320 (Revised).*

**Comment 6**

Section 2.8, please give a definition used in this study.

**Response:** Thanks for the suggestion. We have added explicit definitions of the four forest change modes (forest growth, expansion, loss, and tree cover loss) in Section 2.8 to clarify how they are defined and applied in this study.

*Where modified in manuscript:*

*Methods: Page 10, Lines 280–295.*

*To consistently quantify how forest dynamics influence AGC stocks, we established clear operational definitions of forest change modes based on tree cover transitions. Annual forest cover data from the CATCD were used to determine forest status, with pixels having tree cover $\geq$ 20% classified as forest. In this study, four distinct forest change modes were defined as follows:*

1. *Forest growth: pixels that were forested in both 1985 and 2023 and exhibited an increase in tree cover.*

2. *Tree cover loss: pixels that were forested in both 1985 and 2023 but showed a decrease in tree cover.*

3. *Forest expansion: pixels that transitioned from non-forest in 1985 to forest in 2023.*

4. *Forest loss: pixels that changed from forest in 1985 to non-forest in 2023.*

*For each pixel i, the change in aboveground carbon was calculated as:*

$\Delta AGC_i = AGC_{2023,i} - AGC_{1985,i}$ *(5)*

*Positive $\Delta$AGC values indicate carbon gain, while negative values denote carbon loss. To facilitate interpretation, the four modes were further grouped into two broader categories: 1) Tree cover change (TCC)–induced changes, including forest growth and tree cover loss, which occur without a change in land-cover class; and 2) Land-use and land-cover change (LULCC)–induced changes, including forest expansion and forest loss, which involve transitions between forest and non-forest classes. Finally, we aggregated pixel-level $\Delta$AGC to quantify the relative contributions of each change mode and category to the national AGC balance between 1985 and 2023.*

**Comment 7**

Figure 3, is it make a temporal validation, see comment above

**Response:** We thank the reviewer for this valuable comment. As noted above, following the reviewer's suggestion, we implemented a Leave-One-Year-Out (LOYO) temporal validation following the approach of Laffitte et al. (2025), using the multi-year GEDI AGBD data (2019–2021). Three annual models were trained, each time leaving one year out for validation and repeating the process five times with different random seeds to ensure robustness.